# Travelling spindles create necessary conditions for spike-timing-dependent plasticity in humans

Charles W. Dickey [1,2✉], Anna Sargsyan[3], Joseph R. Madsen[4], Emad N. Eskandar[5], Sydney S. Cash[6,8] & Eric Halgren[3,7,8✉]

Sleep spindles facilitate memory consolidation in the cortex during mammalian non-rapid eye movement sleep. In rodents, phase-locked firing during spindles may facilitate spike-timing-dependent plasticity by grouping pre-then-post-synaptic cell firing within ~25 ms. Currently, microphysiological evidence in humans for conditions conducive for spike-timing-dependent plasticity during spindles is absent. Here, we analyze field potentials and unit firing from middle/upper layers during spindles from 10 × 10 microelectrode arrays at 400 μm pitch in humans. We report strong tonic and phase-locked increases in firing and co-firing within 25 ms during spindles, especially those co-occurring with down-to-upstate transitions. Co-firing, spindle co-occurrence, and spindle coherence are greatest within ~2 mm, and high co-firing of units on different contacts depends on high spindle coherence between those contacts. Spindles propagate at ~0.28 m/s in distinct patterns, with correlated cell co-firing sequences. Spindles hence organize spatiotemporal patterns of neuronal co-firing in ways that may provide pre-conditions for plasticity during non-rapid eye movement sleep.

[1] Neurosciences Graduate Program, University of California, San Diego, La Jolla, CA, USA. [2] Medical Scientist Training Program, University of California, San Diego, La Jolla, CA, USA. [3] Department of Radiology, University of California, San Diego, La Jolla, CA, USA. [4] Division of Epilepsy Surgery, Department of Neurosurgery, Boston Children's Hospital, Harvard Medical School, Boston, MA, USA. [5] Department of Neurological Surgery, Montefiore Medical Center, Albert Einstein College of Medicine, Bronx, NY, USA. [6] Department of Neurology, Massachusetts General Hospital, Harvard Medical School, Boston, MA, USA. [7] Department of Neurosciences, University of California, San Diego, La Jolla, CA, USA. [8] These authors jointly supervised this work: Sydney S. Cash, Eric Halgren. ✉email: cdickey@health.ucsd.edu; ehalgren@health.ucsd.edu

Sleep spindles are bursts of 10–16 Hz oscillations that last for 0.5–2 s at the scalp and occur spontaneously during mammalian non-rapid eye movement (NREM) sleep[1]. It is widely accepted that spindles are important for memory consolidation[2]. Spindles are generated by an interaction of intrinsic currents and local circuits within the thalamus[3,4], and are projected to all cortical areas[5,6]. Spindles often occur on upstates following downstates[6,7].

In the two-stage model of memory, short-term memories are encoded in the hippocampus and then subsequently consolidated into long-term storage by repeated activation of cortical networks during sleep[8,9]. It is hypothesized that hippocampal-to-cortical transfer of memories involves the replay of cell firing sequences during sleep[10], correlated with hippocampal sharp-wave ripples, cortical slow oscillations, and cortical spindles[11–15]. Disrupting this association in rodents impairs consolidation[16], and spindle density is correlated with consolidation in humans[17,18], suggesting that cortical spindles contribute to the consolidation process[19]. However, the mechanisms underlying this phenomenon are not well understood.

It was first proposed that spindles facilitate plasticity by Timofeev[20], who showed that spindles in cats are associated with long-term changes of responsiveness in cortical neurons. Recent studies in rodents have shown that spindles are associated with dendritic $Ca^{2+}$ influxes[21], which may be enhanced through coupling with down-to-upstates[22], and could support plasticity underlying memory consolidation. In support of this hypothesis, in vitro stimulation of rat cortical pyramidal cells pattern matched to in vivo firing sequences during spindles promotes $Ca^{2+}$-dependent long-term potentiation (LTP) of excitatory post-synaptic potentials[23]. Potentiation was dependent on coordinated pre-synaptic potentials and post-synaptic spiking. These observations suggest that spindles may promote spike-timing-dependent plasticity (STDP), which is a $Ca^{2+}$-dependent mechanism where correlated pre- and post-synaptic spiking within a short time window modulates synaptic strength[24]. In the standard model, STDP facilitates LTP when pre-synaptic spiking occurs within 25 ms before post-synaptic spiking, or long-term depression (LTD) when the post-synaptic cell fires first.

Indirect evidence in humans supports the hypothesis that spindles facilitate STDP. Specifically, electrocorticography recordings show that spindles travel across the cortex at ~3–9 m/s, which may be optimal for inducing STDP across distant regions[25]. Furthermore, intracranial studies show that spindle phase modulates high gamma[7,26], which may reflect increased cortical unit co-firing, required for STDP. However, this remains controversial as a previous human intracranial study did not find an increase in unit spiking during spindles[27]. While several limitations preclude the recording of dendritic $Ca^{2+}$ currents in humans, it is possible to test if the unit spike timing requirements for STDP are fulfilled during spindles. Specifically, the most prominent requirement for STDP is co-firing within 25 ms.

Here, we analyzed intracranial microelectrode recordings from a 10 × 10 grid at 400 µm pitch in cortical supragranular layers II/III, and possibly as low as layer IV, in patients undergoing evaluation of pharmaco-resistant intractable epilepsy. We detected putative pyramidal (PY) and interneuron (IN) units and spindles in the local field potential (LFP). Cortical firing was strongly modulated with spindle phase, and increased tonically during spindles, especially when co-occurring with a down-to-upstate. Critically, co-firing within 25 ms of cells recorded by different contacts strongly increased during spindles beyond even what would be expected from the increased firing rate, fulfilling a critical precondition for STDP. Some unit pairs had a preferred order of co-firing, which could support directional plasticity. Spindles tended to co-occur and were highly coherent within ~1.5–2.0 mm, and increased co-firing of units on different contacts was highly enriched when those contacts had highly coherent spindles. Spindles and associated co-firing propagated at ~0.28 m/s with multiple patterns within and between spindles. Thus, spindles spatiotemporally organize neuronal co-firing on a sub-centimeter scale in a manner that could facilitate plasticity across multiple networks.

## Results

### Characterizations of units and spindles.
A mean and standard deviation of 133 ± 50 min of NREM sleep data was selected for analysis from recordings by the Utah Array implanted in supragranular layers II/III, and possibly as low as granular layer IV, of the superior or middle temporal gyrus in 4 patients (Table 1; Fig. 1a, b) with focal epilepsy undergoing monitoring for seizure localization prior to resection. A total of 156 PYs, 39 INs, and 158 multi-units (MUs) were detected, classified, and analyzed (Fig. 1c–f; see Supplementary Fig. 1 for quality and isolation metrics). Therefore, among the single units, 80% were PY and 20% were IN. A total of 340,743 sleep spindles were analyzed across 265 average referenced channels. The mean and standard deviation spindle density per channel was 10.82 ± 3.39 occurrences/minute, duration was 470.32 ± 174.32 ms (Supplementary Fig. 2a), and oscillation frequency was 12.52 ± 1.18 Hz (Supplementary Fig. 2b), which are consistent with previous intracranial studies in humans[6,26]. The mean and standard deviation percent of spindles during which there was at least one spike on the same recording contact from a PY was 7.92 ± 7.33% and from an IN was 36.46 ± 24.31%. The mean and standard deviation percent of co-occurring spindles on different contacts when there was at least one spindle occurring was 29.46 ± 17.11% (Supplementary Fig. 2d).

### Spindles are associated with an increase in unit spike rates.
Prior to testing for co-firing by pairs of units during spindles, our main focus, we characterized neuronal firing during spindles because they provide the context of such co-firing. Specifically, increased firing during spindles, clustered at consistent phases, could result in increased co-firing. Spike rates for each unit were quantified and analyzed during spindles detected on the unit's channel and compared to baseline, which was comprised of randomly selected epochs in between spindles on the same channel that were matched in number and duration to the spindles. PYs and INs increased firing during spindles (Fig. 2a, b). The mean and standard deviation baseline spike rate of PYs was 0.15 ± 0.16 Hz and INs was 1.61 ± 1.63 Hz. The mean and standard deviation spike rate during spindles for PYs was 0.25 ± 0.29 Hz and INs was 2.33 ± 2.35 Hz. There was a significant increase in the mean percent of baseline spike rate during spindles for PYs of 226.97 ± 18.41% (SEM) and for INs of 180.13 ± 21.95% (Fig. 2b, $p_{PY} = 6e–22$, $p_{IN} = 6e–6$, Bonferroni-corrected $\alpha = 0.025$ for 2 unit types, one sample two-sided Wilcoxon signed-rank test, $z_{PY} = 9.64$, $z_{IN} = 4.54$). The increase in spike rates during spindles was present for individual patients (Supplementary Table 1a) and also for shorter (<500 ms) compared to longer (>500 ms) spindles (Supplementary Fig. 3). MU spiking is reported in Supplementary Fig. 4a–d. At shorter distances to spindles, units had higher spike rates, and this fall off across distance was sharper for PYs than INs within the first millimeter (Supplementary Fig. 5).

### Unit spiking phase-locks to the spindle with PY preceding IN.
Grouping of unit spikes by spindle phase could further increase the probability of co-firing between units. PY and IN each had prominent increases in unit spike rates locked to the spindle

**Table 1 Patient demographics, array implantation locations, and unit characteristics.**

**Patient demographics and array implantation**

| Patient | Age | Sex | Handedness | Utah Array implantation location | Probe length (mm) | Recording time (min) |
|---|---|---|---|---|---|---|
| 1 | 51 | F | R | Left middle temporal gyrus | 1.0 | 200 |
| 2 | 31 | M | L | Left superior temporal gyrus | 1.5 | 132 |
| 3 | 47 | M | R | Right middle temporal gyrus | 1.5 | 120 |
| 4 | 21 | M | R | Left middle temporal gyrus | 1.0 | 80 |

**Unit characteristics**

| Unit type | Total units | Total spikes | Valley-to-peak amplitude (µV) | Firing rate (Hz) | Valley-to-peak width (ms) | Half peak width (ms) | Bursting index |
|---|---|---|---|---|---|---|---|
| PY | 156 | 352992 | 84.02 ± 59.45 | 0.19 ± 0.17 | 0.49 ± 0.057 | 0.61 ± 0.038 | 0.046 ± 0.033 |
| IN | 39 | 785571 | 41.60 ± 26.09 | 1.71 ± 1.70 | 0.30 ± 0.054 | 0.35 ± 0.052 | 0.012 ± 0.016 |
| MU | 158 | 5256429 | 26.53 ± 14.89 | 2.82 ± 2.68 | 0.48 ± 0.086 | 0.57 ± 0.070 | 0.046 ± 0.033 |

The total units and total spikes for each unit type across all four patients are reported. The mean and standard deviation across units for all patients for valley-to-peak amplitude, firing rate, valley-to-peak width, half peak width, and bursting index are reported. PY pyramidal unit, IN interneuron unit, MU multi-unit.

trough (Supplementary Fig. 6). The circular mean spindle phase of spikes across the PY ($n = 145$) and IN ($n = 39$) that spiked during spindles was 3.39 rad and 3.64 rad, respectively (Fig. 2c). There was a significant spindle phase preference of 17.24% of PYs and 64.10% of INs ($\alpha = 0.05$, Hodges–Ajne test with bootstrapping). For units with significant spindle phase preferences, the circular mean spindle phase of spikes across PYs was 3.30 rad (Supplementary Fig. 7a) and INs was 3.99 rad (Supplementary Fig. 7b). There was a significant difference between the circular mean spindle phase angle distributions of PY spikes vs. IN spikes, with PY spikes preceding IN spikes ($p = 0.023$, parametric Watson–Williams multi-sample test) for units with a significant phase preference. For 10–16 Hz spindles, this corresponds to a 6.86–10.98 ms delay from PY to IN spiking, and for the mean spindle frequency of 12.52 Hz that we calculated, this corresponds to a 8.77 ms delay. See Supplementary Table 1b for results from individual patients. MU spike-phase results are reported in Supplementary Fig. 4e and Supplementary Fig. 7c.

**Unit spiking increases more during spindles coupled to down-to-upstates.** Spindles often occur during upstates[6], and this coupling appears to be important for memory consolidation[2]. We sought to identify such spindles that coincided with either downstates (which are typically followed by upstates) or upstates (which are typically preceded by downstates) (Supplementary Fig. 8). We found that 45.11% of spindles did not begin within ±1000 ms of downstates or upstates (henceforth referred to as "isolated" spindles), and 7.78% had a downstate peak within 750 ms preceding spindle onset and 6.36% had an upstate peak within 500 ms following spindle onset. The remainder of spindles had downstate or upstate peaks outside of these windows but within ±1000 ms of the spindle (e.g., downstates with peaks following spindle onset) and/or had large positive or negative deflections that were not confirmed as downstates or upstates based on changes in high gamma but were excluded to reduce the chance of contamination of the isolated spindle group by downstates or upstates. There was a significant increase in the baseline spike rate during isolated spindles for PYs of 185.17 ± 11.98% (SEM) and for INs of 150.66 ± 10.35% (Fig. 2d, f; $p_{PY} = 7e-18$, $p_{IN} = 1e-6$, Bonferroni-corrected alpha = 0.025 for 2 unit types, one sample two-sided Wilcoxon signed-rank test, $z_{PY} = 8.61$, $z_{IN} = 4.87$). There was no significant difference for spindles that coincided with downstates vs. those that coincided with upstates ($p_{PY} = 0.11$, $p_{IN} = 0.79$, Bonferroni-corrected $\alpha = 0.025$ for 2 unit types, paired two-sided Wilcoxon signed-rank test, $z_{PY} = -1.61$, $z_{IN} = 0.26$). Therefore, for the remainder of our analyses on spindle interactions with downstates and upstates, we pooled these events into spindles that coincided with down or upstates. There was a significant increase in spindles that coincided with down/upstates for PYs of 243.06 ± 22.66% and for INs of 253.62 ± 56.94% (Fig. 2e, f; $p_{PY} = 4e-11$, $p_{IN} = 4e-5$, Bonferroni-corrected $\alpha = 0.025$ for 2 unit types, one sample two-sided Wilcoxon signed-rank test, $z_{PY} = 6.62$, $z_{IN} = 4.10$; see Supplementary Fig. 9 for separate spindles coinciding with downstates or upstates results). Furthermore, there was a significantly greater increase in PY spiking during spindles that coincided with down/upstates vs. isolated spindles ($p_{PY} = 0.003$, $p_{IN} = 0.22$, Bonferroni-corrected $\alpha = 0.025$ for 2 unit types, paired two-sided Wilcoxon signed-rank test, $z_{PY} = 2.94$, $z_{IN} = 1.23$). Therefore, PY and IN spiking increase during spindles and the increase in PY spiking in particular is enhanced when spindles coincide with down/upstates.

**Spindles group unit pair co-firing within the window of STDP.** Having established that cortical cells fire more during spindles,

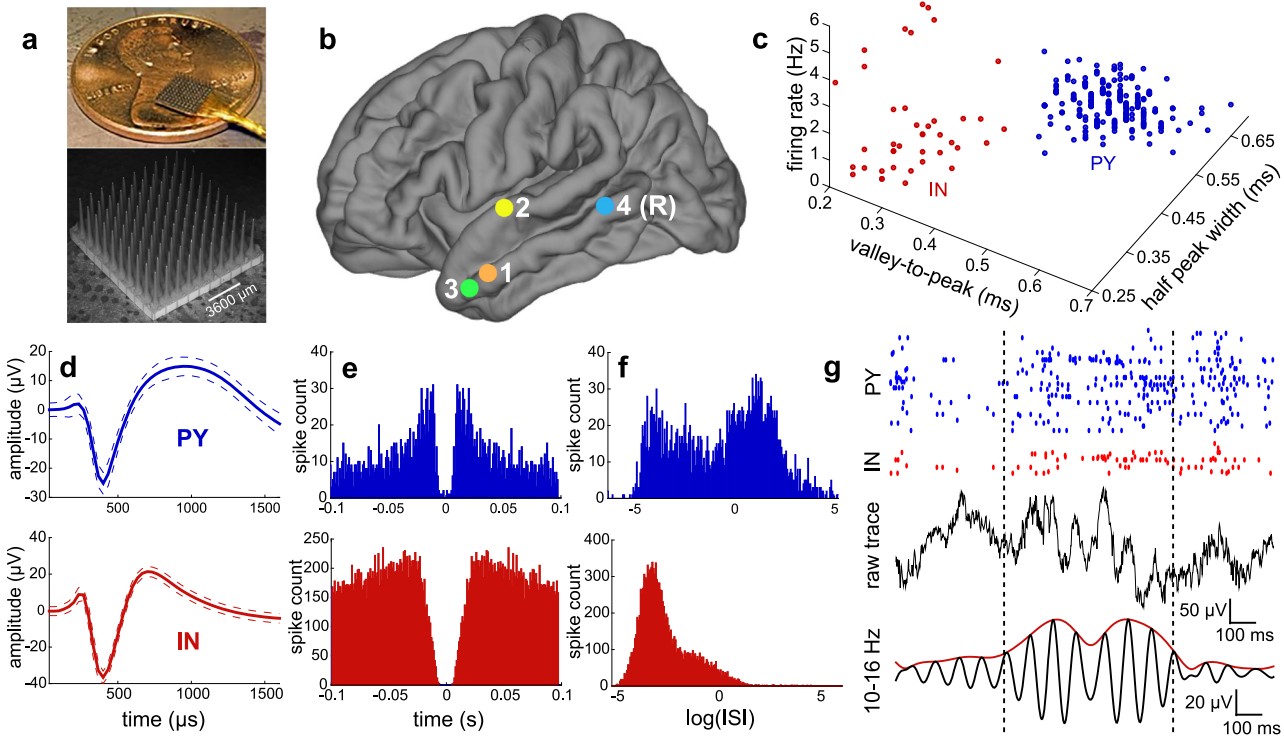

**Fig. 1 Array implantation, single unit classification, spindle detection, and unit spiking during spindles. a** Images of the Utah Array. **b** Utah Array implantation locations for all 4 patients. All sites of implantation were on the left except for patient 4, which was on the right (R). **c** PYs (blue) and INs (red) form separate clusters based on mean waveform half peak width duration, valley-to-peak duration, and firing rate. **d–f** Mean and standard deviation spike waveform (**d**), spike autocorrelation (**e**), and ISI distribution (**f**) for example PY and IN. **g** Raw and 10–16 Hz bandpassed traces of an example spindle with raster plot of associated spiking of an example PY and IN. IN = putative interneuron unit, ISI = inter-spike interval, PY = putative pyramidal unit.

that this firing is phase-locked to individual spindle waves, and is accentuated during down-to-upstates, we turned to our main focus, whether firing is increased within the 25 ms window of STDP during spindles. Unit pair co-firing was quantified by computing peri-spike time occurrences of spikes from all pairs of simultaneously-recorded units, excluding pairs where both units were detected on the same contact. We plotted the spike rate in 1 ms bins across 8026 $PY_1$–$PY_2$ pairs of units during non-spindle epochs (Fig. 3a), spindles (Fig. 3b), isolated spindles that did not coincide with downstates or upstates (Fig. 3c), and spindles that coincided with down or upstates (Fig. 3d), as well as the same for 2127 $PY_1$–$IN_2$ (Fig. 3e–h), 580 $IN_1$–$IN_2$ (Supplementary Fig. 10a–d), and 2127 $IN_1$–$PY_2$ (Supplementary Fig. 10e–h) pairs. For all four types of unit pairs, the spindle distributions were shifted upward and there was a concentrated increase within ~±25 ms for spindle vs. non-spindle epochs, reflecting overall and specific increases in co-firing during spindles. To test the statistical significance of this increase in co-firing within 25 ms, we compared the number of spikes from one unit that occurred within 25 ms before the spikes of another unit for all possible pairs during co-occurring spindles at those two sites individually vs. 1000 sets of randomly selected non-spindle epochs matched in number and duration. There was a significant increase in unit pair co-firing within 25 ms for 16.01% of $PY_1$–$PY_2$, 44.66% of $IN_1$–$IN_2$, 26.28% of $PY_1$–$IN_2$, and 32.96% of $IN_1$–$PY_2$ pairs during spindles vs. non-spindle epochs (α = 0.001, bootstrapped significance, Table 2a). See Supplementary Table 2a for results from individual patients.

**Increased co-firing during spindles is not only due to increased spike rates.** Increased co-firing within the 25 ms window of STDP during spindles could simply be due to the tonic increase in spike

rates during spindles vs. baseline. However, unit co-firing distributions have steeper slopes within 25 ms during spindles (Fig. 3a–h and Supplementary Fig. 10), indicating a specific increase in co-firing. To test if unit pair co-firing increases within 25 ms during spindles independent of firing rates, we compared unit pair co-firing within 25 ms during spindles vs. shuff-spindles (spindles with spike times of each unit randomly shuffled 1000 times). Of note, since the shuffling occurs within spindles, any neuronal drift over time would apply equally to both organized and control spike rates. There was a significant increase in paired co-firing during spindles vs. shuff-spindles for 6.18% of $PY_1$–$PY_2$, 32.41% of $IN_1$–$IN_2$, 7.10% of $PY_1$–$IN_2$, and 11.05% of $IN_1$–$PY_2$ (α = 0.001, bootstrapped significance, Table 2b). About 85% of these unit pairs also increased co-firing significantly during spindles vs. non-spindles (5.41% of $PY_1$–$PY_2$, 25.00% of $IN_1$–$IN_2$, 5.88% of $PY_1$–$IN_2$, and 10.06% of $IN_1$–$PY_2$, α = 0.001, bootstrapped significance, Table 2c). Therefore, the increase in unit pair co-firing during spindles is due not only to the overall firing rate increase, but also to a specific grouping within 25 ms, which is consistent with spindle phase-locked firing. See Supplementary Table 2b, c for results from individual patients.

**Co-firing is greatest during spindles coupled to down-to-upstates.** We next tested whether spindles associated with downstates or upstates had a greater increase in co-firing within 25 ms. We computed the average co-firing rate across unit pairs based on the spikes of $PY_1$ or $IN_1$ within the 25 ms preceding the spikes of $PY_2$ or $IN_2$. The mean and SEM co-firing rate for baseline vs. isolated spindles vs. spindles that coincided with down/upstates for $PY_1$–$PY_2$ was 0.44 ± 0.0072 vs. 0.56 ± 0.029 vs. 0.66 ± 0.063 Hz (Fig. 3a, c, d), for $PY_1$-$IN_2$ was 2.76 ± 0.071 vs. 3.27 ± 0.15 vs. 4.60 ± 0.32 Hz (Fig. 3e, g, h). The co-firing rate

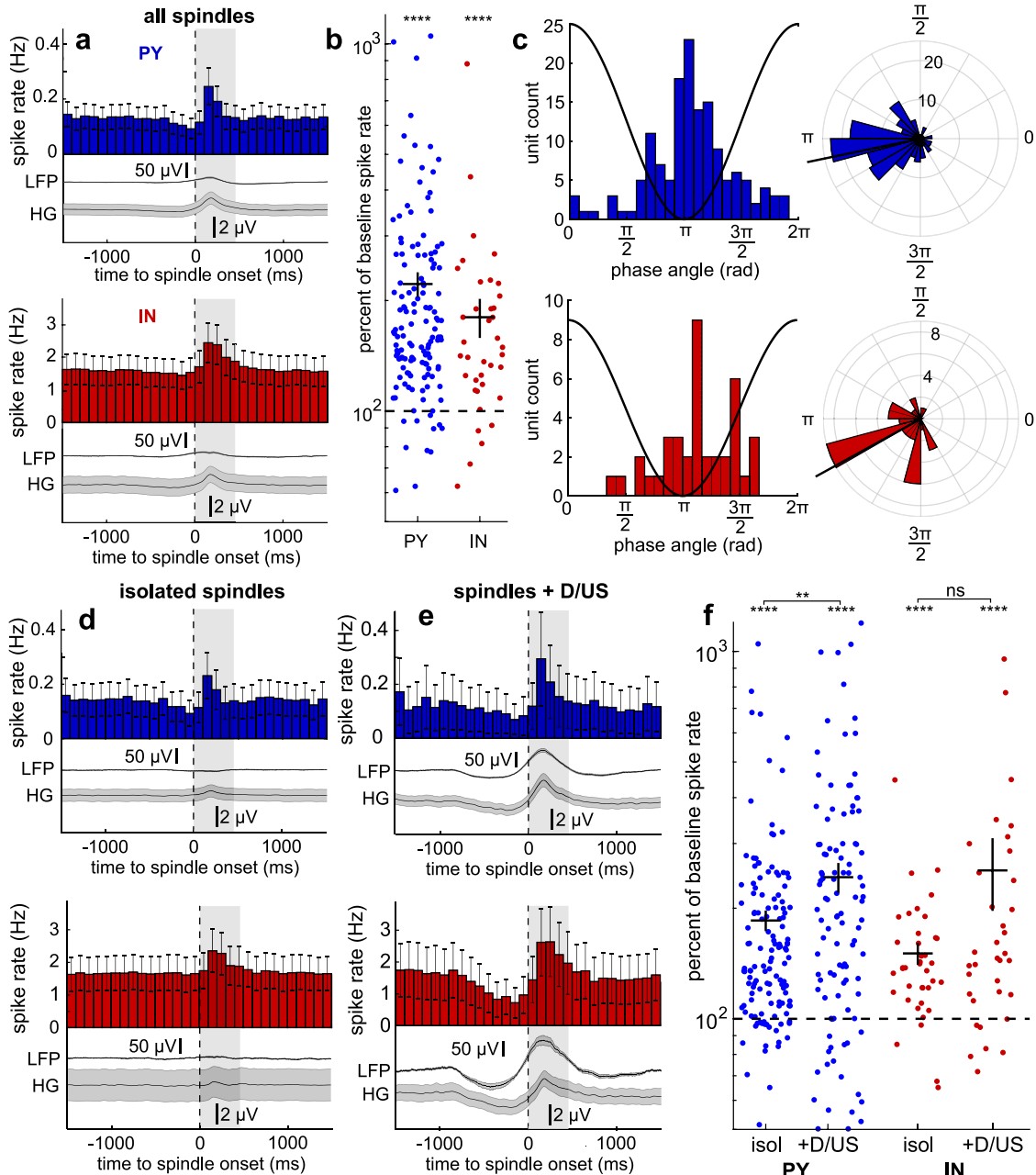

**Fig. 2 Single unit spiking during spindles. a** Mean PY ($n = 156$) and IN ($n = 39$) spike rates and associated mean LFP and HG envelope locked to spindle onsets at $t = 0$. Shaded gray box shows the mean spindle duration of 470 ms. **b** PY ($n = 156$) and IN ($n = 39$) spike rates during concatenated spindle epochs as a log-percent of baseline spike rates ($p_{PY} = 6e{-}22$, $p_{IN} = 6e{-}6$). Baseline spike rate of each unit was computed as the spike rate during concatenated randomly selected NREM sleep epochs between spindles that were matched in number and duration. Color circles represent units. Solid horizontal lines show mean and vertical show SEM. Dashed horizontal line shows baseline spike rate (100%). **c** Non-polar and polar histograms show circular mean spindle phases of spiking for PYs and INs. One cycle of a spindle is superimposed on non-polar histograms to visualize the phase-spike timing relationship. Black lines extending from polar histograms show circular means. **d**, **e** Same as (**a**) except for isolated spindles (**d**), which were those that did not coincide with down/upstates, and spindles that coincided with down/upstates (**e**). **f** Same as (**b**) except for isolated spindles ("isol"; $p_{PY} = 7e{-}18$, $p_{IN} = 1e{-}6$ for spiking compared to baseline) and spindles that coincided with down or upstates ("+D/US") ($p_{PY} = 4e{-}11$, $p_{IN} = 4e{-}5$ for spiking compared to baseline; $p_{PY} = 0.003$, $p_{IN} = 0.22$ for isol vs. +D/US). HG = high gamma, IN = putative interneuron unit, LFP = local field potential, NREM = non-rapid eye movement, PY = putative pyramidal unit. Error bars and shaded errors show SEM. **p < 0.01, ****p < 0.0001, one sample two-sided Wilcoxon signed-rank test for comparisons with baseline, paired two-sided Wilcoxon signed-rank test for isol vs. +D/US.

during isolated spindles was significantly higher than non-spindles ($p_{PY{-}PY} \approx 0$, $p_{PY{-}IN} = 9e{-}33$, Bonferroni-corrected $\alpha = 0.025$ for 2 spindle types, paired two-sided Wilcoxon signed-rank test, $z_{PY{-}PY} = 41.93$, $z_{PY{-}IN} = 11.92$), and the co-firing rate during spindles that coincided with down/upstates was significantly higher than during isolated spindles ($p_{PY{-}PY} = 8e$ $-24$, $p_{PY{-}IN} = 5e{-}11$, Bonferroni-corrected $\alpha = 0.025$ for 2 spindle types, paired two-sided Wilcoxon signed-rank test, $z_{PY{-}PY} = 10.07$, $z_{PY{-}IN} = 6.57$). Therefore, there is an increase in unit pair co-firing during spindles that is enhanced when spindles coincide with down-to-upstates. Results for $IN_1{-}IN_2$ and $IN_1{-}PY_2$ are reported in Supplementary Fig. 10.

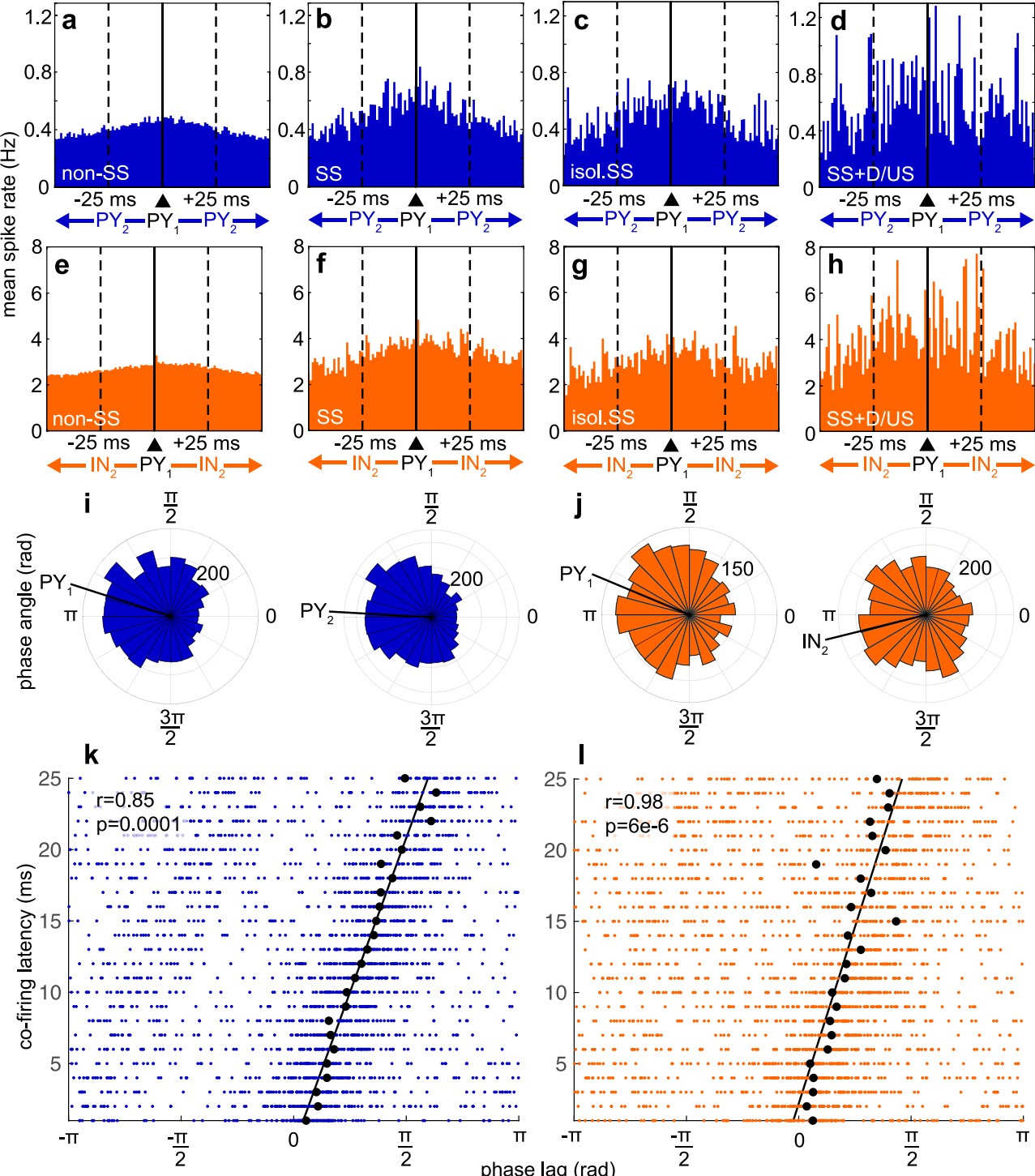

**Fig. 3 Unit pair co-firing during spindles. a–d** In all pairs of PY recorded from different contacts, the firing of one ($PY_2$) is plotted relative to the time of the other ($PY_1$, at $t = 0$) during non-spindle (baseline) epochs (**a**), co-occurring spindles (**b**), isolated co-occurring spindles that did not coincide with down/upstates (**c**), and co-occurring spindles that coincided with down/upstates (**d**). Solid vertical line shows $t = 0$. Dashed vertical lines show the ±25 ms interval where paired pre- and post-synaptic spiking facilitates STDP. **e–h** Same as (**a–d**), except for $PY_1$–$IN_2$. **i** Polar histograms showing spindle phases of $PY_1$ and $PY_2$ spikes when there was co-located co-firing during co-occurring spindles within 25 ms. Phases are according to the spindle detected on the same channel as $PY_1$. Black lines show circular means. **j** Same as (**i**) except for $PY_1$–$IN_2$. **k** Co-firing within 25 ms as a function of spindle phase lag. Co-located unit pair co-firing was identified during co-occurring spindles, and the phase lag of the spindle was computed between the spindles on the two channels co-located with the two units. Black circles indicate circular means for each 1 ms binned latency and black line indicates circular-linear best fit of these means. *P*-value reports the significance of the correlation coefficient. **l** Same as (**k**) except for $PY_1$–$IN_2$.

**Table 2 Significance of paired and ordered unit co-firing.**

| Unit pair (1→2) | a. Paired co-firing: spindles vs. non-spindles | b. Paired co-firing: spindles vs. shuff-spindles | c. Paired co-firing: both | d. Ordered co-firing |
|---|---|---|---|---|
| $PY_1$–$PY_2$ | 16.01% | 6.18% | 5.41% | 19.15% |
| $IN_1$–$IN_2$ | 44.66% | 32.41% | 25.00% | 15.52% |
| $PY_1$–$IN_2$ | 26.28% | 7.10% | 5.88% | 16.67% |
| $IN_1$–$PY_2$ | 32.96% | 11.05% | 10.06% | 21.31% |

**a–c** Percent of unit pairs with significantly increased ($\alpha = 0.001$, bootstrapped significance) co-firing within 25 ms during spindles vs. non-spindles (**a**), spindles vs. shuff-spindles (**b**), and both (**c**).
**d** Percent of unit pairs in (**c**), which also had a minimum number of co-firing events of 10, with significant order preference of co-firing within 25 ms ($\alpha = 0.05$. two-sided $\chi^2$ test of proportions).
Shuff-spindles = spindles with spike times randomly shuffled 1000 times.

**Unit pairs have ordered co-firing during spindles**. In the canonical model, STDP is an order-dependent process that can lead to LTP or LTD[24]. We first tested whether the unit pairs with significant co-firing were significant in one direction or both directions. Among pair sets that were at least significant in one direction, about 15–25% of $PY_1$–$PY_2$ and 55–65% of $IN_1$–$IN_2$ were significant in both directions (Supplementary Table 3). Next, we tested whether unit pairs with significantly increased co-firing within 25 ms for both spindles vs. non-spindles and spindles vs. shuff-spindles had a preferred order of firing within this window. Of 94 $PY_1$–$PY_2$, 116 $IN_1$–$IN_2$, 72 $PY_1$–$IN_2$, and 183 $IN_1$–$PY_2$ pairs, which were only those with ≥10 co-firing spikes within ±25 ms, 19.15% of $PY_1$–$PY_2$, 15.52% of $IN_1$–$IN_2$, 16.67% of $PY_1$–$IN_2$, and 21.31% of $IN_1$–$PY_2$ had a preferred order of co-firing within 25 ms ($\alpha = 0.05$, two-sided $\chi^2$ test of proportions, Table 2d). See Supplementary Table 2d for results from individual patients.

**Co-firing delays are correlated with spindle phase lags**. Since unit spiking is locked to spindle phase and there is an increase in co-firing during spindles, we tested whether co-firing was also locked to spindle phase. For $PY_1$–$PY_2$, when $PY_2$ fired within 25 ms following $PY_1$, $PY_1$ spikes had a circular mean spindle phase of 2.83 rad and $PY_2$ spikes had a circular mean phase of 3.07 rad (Fig. 3i), as measured according to the phases of the spindle co-located with $PY_1$. Likewise for $PY_1$–$IN_2$, $PY_1$ had a circular mean phase of 2.75 rad and $IN_2$ had a circular mean phase of 3.22 rad (Fig. 3j). There was no significant difference between spindle phase of $PY_1$ in $PY_1$–$PY_2$ vs. $PY_1$ in $PY_1$–$IN_2$, or between spindle phase of $PY_2$ in $PY_1$–$PY_2$ vs. $IN_2$ in $PY_1$–$IN_2$ ($p = 0.11$ and $p = 0.91$, respectively, Bonferroni-corrected $\alpha = 0.025$ for 2 pair types, parametric Watson–Williams multi-sample test). When we analyzed co-firing delay vs. circular mean spindle phase lag, computed between the spindles co-located with each unit, there was a significant circular-linear relationship for both $PY_1$–$PY_2$ (Fig. 3k, $r = 0.85$, $p = 0.0001$, upper tail probability of the $\chi^2$ distribution) and $PY_1$–$IN_2$ (Fig. 3l, $r = 0.98$, $p = 6e{-}6$, upper tail probability of the $\chi^2$ distribution). Thus, when two cells recorded by different contacts fire within 25 ms of each other during spindles, the latency between their spikes is highly correlated with the phase lag between the spindles recorded by the two contacts.

**Increased co-firing during spindles is enhanced at short distances**. The proportion of unit pairs with significantly increased co-firing within 25 ms during spindles vs. shuff-spindles had a significant negative linear relationship to contact separation for $PY_1$–$PY_2$ (Fig. 4a, $r = -0.39$, $p = 0.02$, significance of the correlation coefficient), indicating that at shorter distances there is more organized 25 ms co-firing. The increased tendency of cortical neurons to fire within 25 ms of each other was confirmed with the spike time tiling coefficient[28], an alternative analysis

method that is independent of firing rates, for both $PY_1$–$PY_2$ and $IN_1$–$IN_2$. This method also revealed a similar drop-off in co-firing with distance (Supplementary Fig. 11). Based on the relationship between co-firing and distance (Fig. 4a), as well as the previously reported neuron densities in human anterior temporal lobe[29], we estimated that the number of layer III PY that co-fire more during spindles with a given PY within a radius of 4 mm, beyond what would be expected from a simple increase in firing rate, was 37,532 (see "Methods").

**Spindles propagate across the microarray at a characteristic velocity**. Spindles have previously been shown to propagate across the cortex on a macro-scale, which was proposed as a mechanism for facilitating STDP between distant cortical regions[25]. The velocity of a traveling wave that oscillates at a known frequency can be calculated based on its phase lag across distance. We found a significant positive linear relationship between distance and the magnitude of the spindle phase lag between co-occurring spindles (Fig. 4b, $r = 0.996$, $p = 3e{-}43$, significance of the correlation coefficient). This strong linear relationship indicates unified spindling across multiple contacts of the array. Based on the magnitude of phase lag between co-occurring spindles as a function of distance, we estimated the spindle propagation velocity using the slope of the equation of the linear least squares regression, $y = 2.82e{-}4x + 0.28$, and the mean spindle frequency of 12.52 Hz (see "Methods"). This calculation yielded a spindle propagation velocity of 0.28 m/s. Of note, the phase lag between spindles recorded by different contacts is not consistent with the recording of a common generator that volume conducts between contacts because volume conduction is effectively instantaneous.

**Co-firing is correlated with spindle coherence**. Spindle co-occurrence density (the rate of spindle co-occurrence between two contacts) was greatest at the smallest inter-contact distance of 400 µm and decreased sharply until ~1000 µm, and then plateaued up to the maximum inter-contact distance of ~4000 µm (Fig. 4c; $R^2 = 0.98$, two term exponential least squares regression). The spindle co-occurrence density was greater than chance at all distances (Bonferroni-corrected $\alpha = 0.001$ for 35 distance bins, mean $p = 2e{-}32$, range $= 8e{-}120{-}7e{-}31$, paired two-sided $t$-test, mean $t = 22.43$, range $= 16.13{-}30.28$, mean Cohen's $d = 1.46$, range $= 1.19{-}1.91$), when the chance spindle co-occurrence density was computed by randomly shuffling the inter-spindle intervals 100 times for each channel of each channel pair and then finding the mean co-occurrence density. We next tested if coherence within the 10–16 Hz band between contacts during co-occurring spindles was associated with co-firing recorded by those contacts. Coherence was used to evaluate the relationship between spindles because it depends on both the phases and co-amplitude of signals, which are relevant to the underlying neuronal activity. The magnitude squared coherence of focal LFP within the 10–16 Hz band during spindles was greater at all

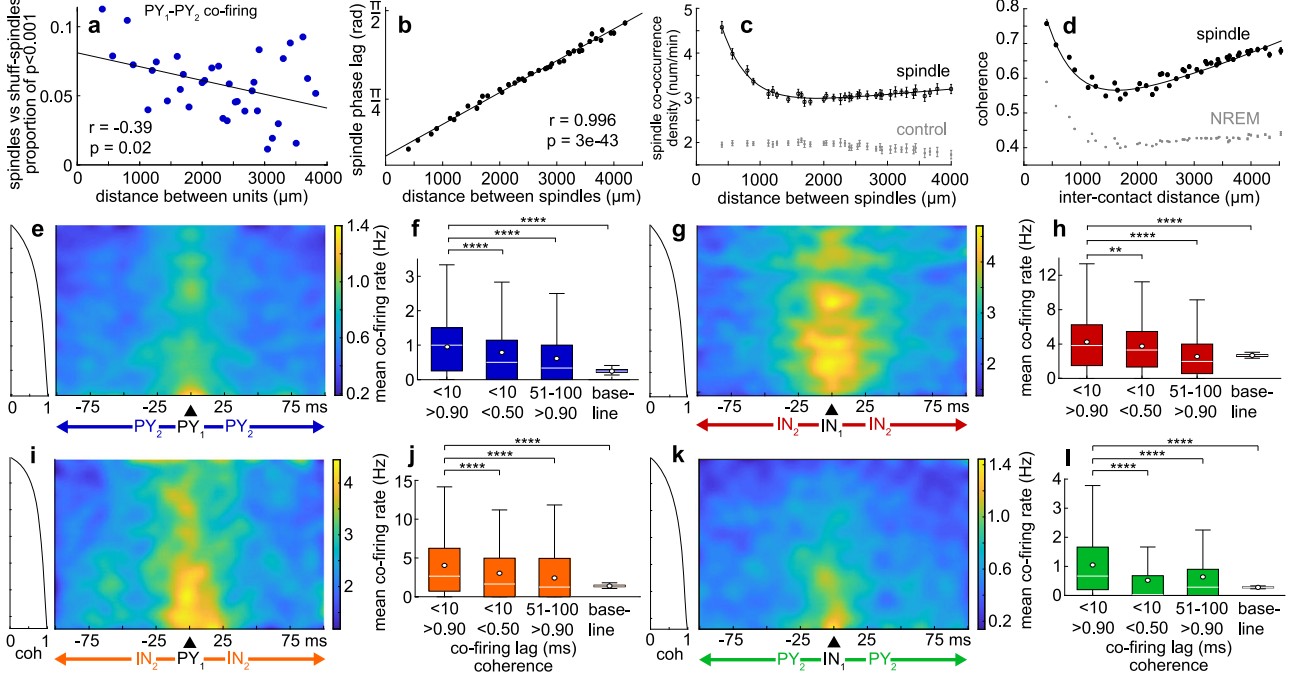

**Fig. 4 Spindle coherence and unit pair co-firing. a** Mean proportion of unit pairs with significant co-firing within 25 ms for spindles vs. shuff-spindles over inter-contact distance for $PY_1$–$PY_2$. Shuff-spindles = spindles with spike times randomly shuffled 1000 times. **b** Magnitude of the phase lag between co-occurring spindles as a function of distance ($n = 1000$ minimum spindle pairs per bin). P-values in **a**, **b** report the significance of the correlation coefficient. **c** Density of co-occurrence of spindle pairs (black) and spindles with inter-spindle intervals randomly shuffled 100 times for each channel of each channel pair (gray) over distance ($n = 100$ minimum channel pairs per bin). **d** Magnitude squared coherence in the 10–16 Hz band during co-occurring spindles (black) and during randomly selected NREM epochs matched in number and duration (gray) over distance ($n = 100$ minimum channel pairs per bin). **e** $PY_1$–$PY_2$ heatmaps of unit pair co-firing rates in pseudo-color as a function of co-firing delay on the abscissa and spindle-spindle coherence within the 10–16 Hz band on the ordinate. Co-firing values are in 1 ms bins and coherence values are binned with a minimum number of spindles per bin (see "Methods") with the coherence indicated in the plot on the left (coherence increases non-linearly from 0 to 1 from north to south). Heatmaps were smoothed with a 2D Gaussian filter with $\alpha = 5$. **f** Mean co-firing rates were quantified from the data in (**e**) within heatmap regions with shorter co-firing delays (<10 ms) and higher coherence (>0.90) in the left distribution ($n = 777$ bins), compared to shorter delays and lower coherence (<0.50) ($p = 3\mathrm{e}{-4}$, $n = 525$ bins), longer delays (51–100 ms) and higher coherence ($p = 4\mathrm{e}{-39}$, $n = 4736$ bins), and longer delays and lower coherence during baseline NREM periods in between spindles ($p = 9\mathrm{e}{-17}$, $n = 102$ bins). **g–l** Same as (**e**, **f**), except for $IN_1$–$IN_2$ (**g**, **h**) (first comparison: $p = 0.004$, $n = 798$ and $n = 651$ bins; second comparison: $p = 6\mathrm{e}{-46}$, $n = 798$ and $n = 4864$ bins; third comparison: $p = 5\mathrm{e}{-6}$, $n = 798$ and $n = 102$ bins), $PY_1$–$IN_2$ (**i**, **j**) (first comparison: $p = 8\mathrm{e}{-5}$, $n = 861$ and $n = 315$ bins; second comparison: $p = 1\mathrm{e}{-43}$, $n = 861$ and $n = 5248$ bins; third comparison: $p = 2\mathrm{e}{-11}$, $n = 861$ and $n = 102$ bins), and $IN_1$–$PY_2$ (**k**, **l**) (first comparison: $p = 3\mathrm{e}{-16}$, $n = 756$ and $n = 441$ bins; second comparison: $p = 4\mathrm{e}{-29}$, $n = 756$ and $n = 4608$ bins; third comparison: $p = 4\mathrm{e}{-11}$, $n = 756$ and $n = 102$ bins). Boxplot central line shows median, circle shows mean, box limits show lower and upper quartiles, whiskers show 1.5 × interquartile range, and outliers are not depicted. Heatmaps were smoothed with a 2D Gaussian filter with $\alpha = 5$. **\*\***$p < 0.01$, **\*\*\*\***$p < 0.0001$, Bonferroni-corrected $\alpha = 0.017$ for 3 comparisons of delay/coherence, two sample two-sided t-test.

distances compared to randomly selected NREM epochs matched in number and duration (Fig. 4d; Bonferroni-corrected $\alpha = 0.001$ for 47 distance bins, mean $p \approx 0$, range $\approx 0$, two sample two-sided t-test, mean $t = 98.79$, range $= 28.30$–$190.02$, mean Cohen's $d = 0.51$, range $= 0.35$–$0.71$). High levels of co-firing between units recorded by different contacts were restricted to high levels of spindle coherence (>~0.90) between those contacts during co-occurring spindles (Fig. 4e–l). This co-firing was concentrated at short delays (<~10 ms), and was observed for all unit pair types. The mean $PY_1$–$PY_2$ < 10 ms co-firing rate was significantly higher for high (>0.90) vs. lower (<0.50) spindle coherence (Fig. 4f; $p = 3\mathrm{e}{-4}$, Bonferroni-corrected $\alpha = 0.017$ for 3 combinations of delay/coherence, two sample two-sided t-test, $t = 3.67$, Cohen's $d = 0.21$). The mean $PY_1$–$PY_2$ co-firing rate was also significantly higher during spindles with high coherence and short (<10 ms) vs. longer (51–100 ms) co-firing delays ($p = 4\mathrm{e}{-39}$, Bonferroni-corrected $\alpha = 0.017$ for 3 comparisons of delay/coherence, two sample two-sided t-test, $t = 13.19$, Cohen's $d = 0.51$). The mean $PY_1$–$PY_2$ co-firing rate increased by ~385% compared to baseline NREM periods between spindles with longer delays and lower coherence ($p = 9\mathrm{e}{-17}$, Bonferroni-corrected

$\alpha = 0.017$ for 3 combinations of delay/coherence, two sample two-sided t-test, $t = 8.48$, Cohen's $d = 0.89$). See Supplementary Table 4 for additional details. Thus, short latency unit co-firing depends critically on spindle coherence being close to 1.

**Spindles travel across the microarray in multiple patterns.** Spindle propagation could facilitate sequential neuronal co-firing events leading to patterned synaptic strengthening within a network. Different waves within an individual spindle were capable of exhibiting different patterns of propagation (Fig. 5; Supplementary Movie 1). For example, one spindle wave had a circular propagation pattern, based on its z-score normalized amplitude (Fig. 5a, e) and phase (Fig. 5b, f), and in subsequent wave cycles showed a planar propagation pattern (Fig. 5c–f). We used the MATLAB: NeuroPatt Toolbox[30] (see "Methods") to find spatio-temporal modes, represented as phase velocity vector fields, that explained the greatest percent variance of the phase velocity vector time series for each spindle (Fig. 6). Controls were generated by randomly shuffling the positions of the good channels

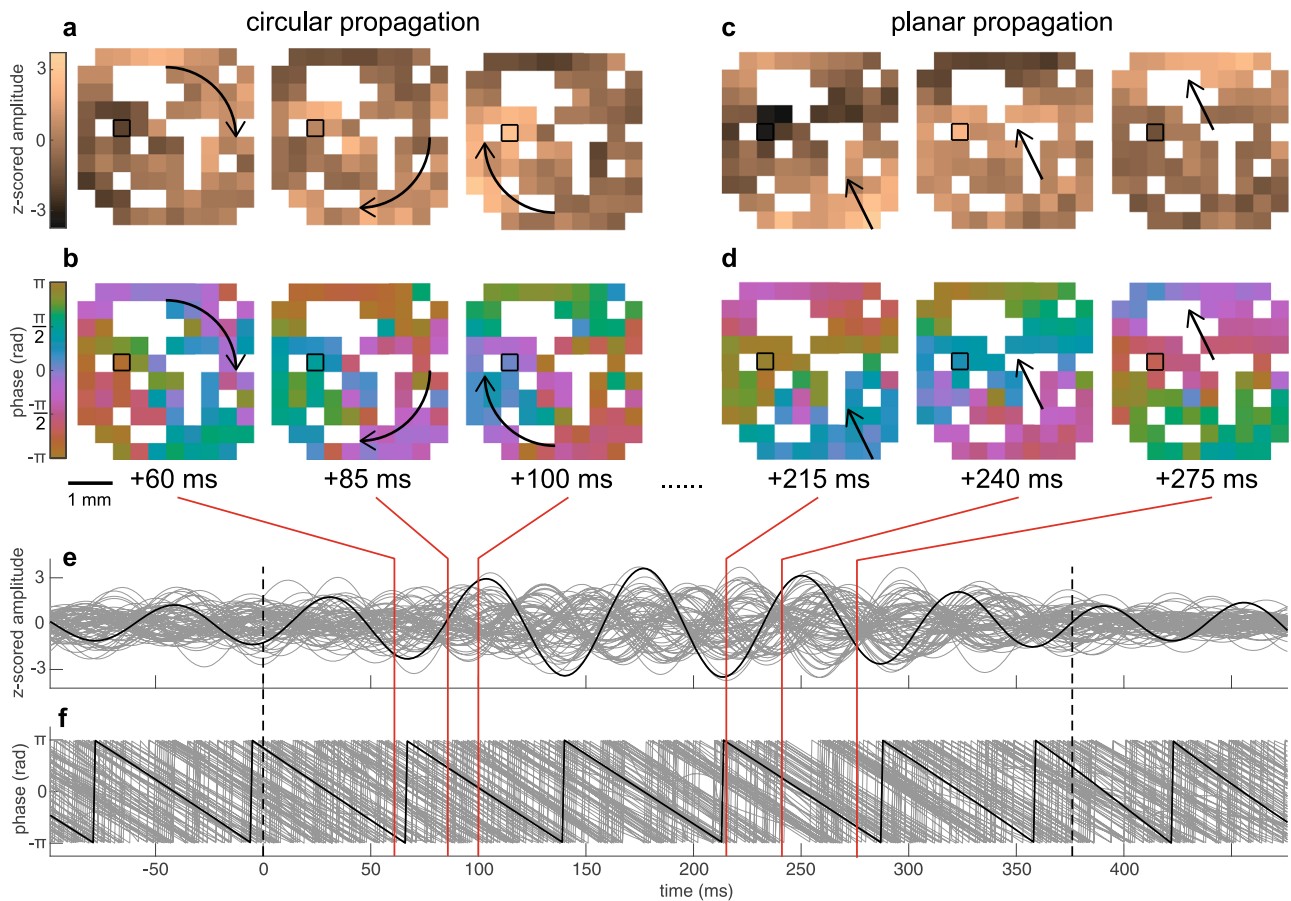

**Fig. 5 Spindles propagate on a sub-centimeter scale. a, b** Circular propagation of a spindle wave depicted in 3 frames of the instantaneous z-score normalized amplitude (**a**) and instantaneous phase (**b**). A cyclic colormap was used to show phase. **c, d** Same as (**a, b**) but for linear propagation for a different wave within the same spindle. White spaces indicate non-existent or bad channels. The channel outlined in black corresponds to the channel on which the spindle was detected. Arrows indicate approximate trajectory of propagation. **e, f** Traces of z-score normalized amplitude (**e**) and phase (**f**) of the same spindle in (**a–d**). Black trace corresponds to the channel on which the spindle was detected and gray traces show the rest of the channels. Red lines extending from (**e, f**) show the times of the frames in (**a–d**). See Supplementary Movie 1.

once for each spindle prior to interpolation and spatiotemporal analysis. The percent explained variances of modes 1 and 2, i.e., those with the greatest percent explained variance, were greater for spindles (mean and SEM for mode 1 was $22.42 \pm 0.15\%$ and for mode 2 was $14.67 \pm 0.06\%$) vs. controls (mode 1 was $11.99 \pm 0.04\%$ and mode 2 was $10.20 \pm 0.03\%$) in all subjects (Fig. 6a, b; mode 1: $p \approx 0$, $t = 47.86$, Cohen's $d = 0.93$; and mode 2: $p \approx 0$, $t = 36.16$, Cohen's $d = 0.73$, paired two-sided $t$-test), providing confirmation of propagating spindles. There were a variety of propagation patterns within and across patients and spindles (representative examples of mode 1 in Fig. 6c), demonstrating that spindles have multiple patterns of propagation within $3.6 \times 3.6$ mm of cortex.

**Distinct co-firing and spindle propagation patterns are linked.** The above spindle propagation analysis was performed for each spindle independently, and so the patterns could also change across spindles. In order to analyze the relationship to co-firing it was necessary to identify consistent propagation patterns. Thus, a second analysis was performed on a concatenation of all spindles from each patient. Again, distinct modes were consistently found as in Fig. 6c. $PY_1$–$PY_2$ co-firing was then quantified separately for modes 1 and 2 of the concatenated spindles for each patient when singular value decomposition component scores exceeded the 75th percentile, representing periods when each mode was most prominent. Among 187 pairs where $PY_2$ co-fired within 25 ms

following $PY_1$ at least 10 times between modes 1 and 2, 177 pairs (94.65%) had a significant difference in co-firing for modes 1 vs. 2 ($\alpha = 0.05$, two-sided $\chi^2$ test of proportions). Thus, different patterns of plasticity may be induced by different patterns of spindle propagation.

## Discussion

We identified the spatiotemporally-patterned inter-relations of LFPs and neuronal firing during human sleep spindles over a $10 \times 10$ microelectrode array with 400 µm pitch. Firing of individual putative cortical PYs and INs increased during spindles, with an additional increase at certain spindle phases. Co-firing within 25 ms between neurons recorded on different micro-contacts, a pre-requisite for STDP, also increased during spindles in a phase-locked manner. Co-firing of cells and co-occurrence of spindles were greatest at inter-contact separations <1 mm but extended over the entire array. Both firing and co-firing were greatest when spindles occurred on down-to-upstates, which is consistent with the proposition that spindles that occur on this transition are especially important for memory consolidation[16]. The mean co-firing delay between cells and phase lag between coherent spindles increased linearly with distance. Conduction speed and PY-IN phase relations were consistent with direct cortico-cortical spindle propagation. Short-latency co-firing was concentrated between contact pairs with highly coherent spindles. Multiple two-dimensional spindle propagation patterns and

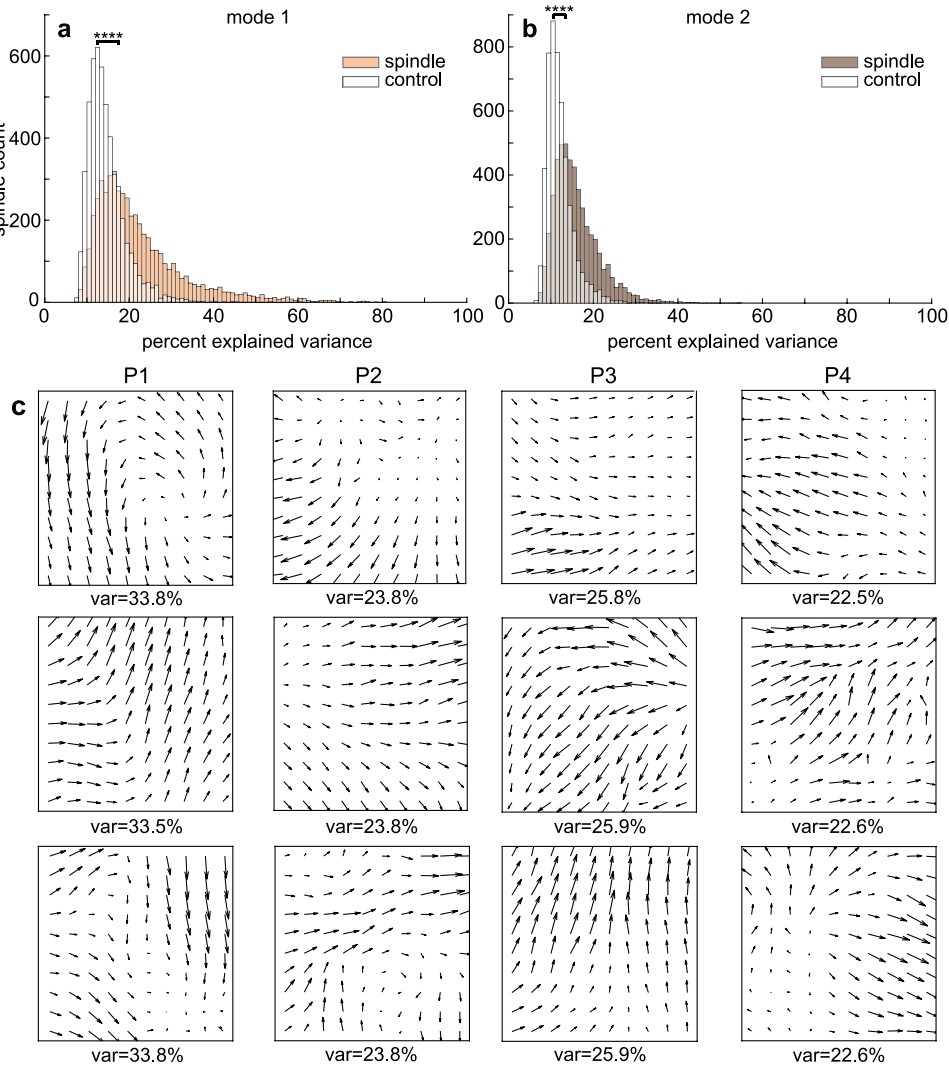

**Fig. 6 Spatiotemporal propagation patterns of spindles. a** Percent explained variance (var) for most dominant spatiotemporal mode (mode 1) across spindles (tan) and controls (white) where good channel positions were randomly shuffled once per spindle prior to interpolation ($p \approx 0$). ****$p < 0.0001$, paired two-sided $t$-test. **b** Same as (**a**) except for mode 2 ($p \approx 0$). **c** Three representative examples of mode 1 phase velocity vector fields showing individual spindle propagation patterns for each patient (P1–4).

associated distinct co-firing patterns occurred across each array, intermixed within and between spindles. Overall, these micro-physiological mechanisms may support and organize memory consolidation by creating the necessary conditions for STDP and activating spatiotemporal networks through travelling spindles.

We found that part of the increase in co-firing during spindles is due to the two-fold increase in firing they induce. This finding was in contrast to Andrillon et al.[27] who found no increase in firing during spindles in humans. This difference could be because they sampled medial limbic cortex whereas we sampled lateral temporal cortex. Furthermore, they correlated unit spiking detected by microwires with LFP recorded by a macroelectrode contact ~4 mm away, whereas we correlated unit spiking with LFP recorded from the same contacts, and they recorded from all cortical layers whereas we only recorded from supragranular and possibly granular layers. Indeed, the increase in unit firing during spindles was ~50% smaller at a distance of ~4 mm, and it has been previously shown that spindle-phase modulation of high gamma in humans and of unit firing in rodents is greater in supragranular vs. infragranular layers[26,31]. Thus, our recordings focused on the most responsive layers. We furthermore found that PY and IN spiking was locked to the phase of individual

spindle waves, which is consistent with what has been previously shown for PY and IN in rodents[31] and for units without cell type classification in humans[27], and suggests one mechanism whereby spindles specifically coordinate co-firing beyond a mere general tonic increase in firing.

Our study provides the first direct evidence that the essential pre-condition for STDP, unit pair co-firing within 25 ms, is met in the human cortex during spindles in NREM sleep. This finding helps resolve a major issue in understanding how STDP could occur under normal conditions: the large number of repetitions, up to hundreds, that are needed to produce long-term changes[32]. Co-firing during spindles could address this requirement because in a normal night's sleep, ~1000 spindles will occur at most cortical sites, and each spindle has ~10 cycles, so there are ample opportunities for STDP repetitions. Furthermore, spindle frequency lies within the repetition rate for which such pairings are effective[24]. In addition, memory-related cortical input from the hippocampus associated with ripples may be available on multiple spindle peaks often seen in the posterior hippocampus, phase-locked with cortical spindles[12,33]. Of note, all of the recordings in this study were from the anterolateral temporal cortex, which is not part of the medial prefrontal cortex-hippocampal network

that is typically associated with sleep-dependent memory consolidation in rodents. However, rodents have no clear homolog to the anterolateral temporal cortex, which in humans is thought to be a key part of the memory system[34,35].

In canonical $PY_1$–$PY_2$ STDP, pre-before-postsynaptic spiking leads to LTP and the reverse leads to LTD[24]. Plasticity underlying memory consolidation may involve both[36]. Increased co-firing within 25 ms for all combinations of PY and IN was found during co-occurring spindles over 4 mm, suggesting an extensive network of co-firing cells. Based on neuronal density in the human cortex, and the co-firing probability-distance function we observed, it can be roughly estimated that the number of PY that co-fire more during spindles with a given PY within a radius of 4 mm, beyond what would be expected from a simple increase in firing rate, is ~37,500 for layer III alone. Many of the pairs with increased co-firing had a preferred order of spiking, which could support unidirectional plasticity. However, we did not explicitly test whether the units modulated during spindles underwent synaptic weight changes or were involved in plasticity underlying memory consolidation, which should be assessed in future work. Human slice recordings have shown that local excitatory connectivity between layers II/III PYs is 13–18%[37], substantially higher than in mice[38]. However, most co-firing within 25 ms is probably by unit pairs that are not directly connected but belong to the same local network. In some cases this is directional, however, there could be multiple pathways between co-firing units, some in one direction and others in the opposite direction, and network tuning would involve strengthening some routes and weakening others. In any case, the network of co-firing cells appears to be extensive.

Human magnetoencephalography[39] and intracranial macro-electrode[5,27] recordings have shown that spindles, once thought to be a global phenomenon, are often focal at a centimeter scale. Human laminar recordings have furthermore shown that spindles localize to specific cortical layers[26,40], however the lateral extent of spindles in human cortex has not been reported on a sub-centimeter scale. We show that spindle co-occurrence and coherence in human cortex peaks at the shortest inter-contact distance of 400 μm, decreasing sharply to a plateau at ~1000 μm. Since an average reference would eliminate spindles that are equal across all leads, there may also be co-occurrence at a larger scale, and indeed, asymptotic co-occurrence and coherence exceeded chance. Taken together, the data indicate that the cortical extent engaged by spindles can range from a few columns to much of the cortex.

Traveling waves may gate the flow of spiking in cortical circuits[41]. We found that spindles propagated within the microgrid at ~0.28 m/s. This is within the range of or slightly lower than previously reported intracortical axon conduction velocities, including for layers II/III, of 0.28 m/s and 0.15–0.44 m/s in rat visual cortex[42,43], 0.35–0.45 m/s in rat neocortex[44], and 0.35 m/s in cat visual cortex[45]. The true axonal conduction velocity of spindles may be faster than 0.28 m/s because our calculation assumes a direct path of travel and does not take synaptic delays into account. This velocity is much slower than what was reported by Muller et al.[25] for human cortical spindles (3–9 m/s) and Halgren et al.[46] for human cortical alpha (0.91 m/s), both using ECoG recordings, presumably because they were measuring fast conduction via myelinated fibers passing through the white matter, whereas we were measuring slow conduction via unmyelinated fibers within the cortical gray matter. The different velocities may promote plasticity in networks of different spatial extents[25,47].

Direct cortico-cortical propagation of spindles is at odds with the common conception of cortical spindles being driven from the thalamus. In cats, the thalamus continues to spindle after cortical removal, but the cortex does not spindle after disconnection from the thalamus[48]. In mice, rhythmic optogenetic activation of the thalamic reticular nucleus triggers spindles[49]. In humans, thalamic spindles occur more frequently and begin before cortical, and in rare cases show tight phase-locking with thalamus leading the cortex[6]. Thus, spindles are thought to originate thalamically and project cortically[50]. Thalamocortical projections in mice to both primary sensory and limbic cortices drive INs more strongly and at shorter latencies than PYs[51,52]. Thus, our finding that PY spiking precedes IN spiking is not consistent with cortical spindles in humans being mainly driven by the thalamus. Rather, it is possible that while cortical spindles in humans are initially driven by the thalamus, intrinsic cortical circuits may subsequently amplify and spread the spindle. Local generation seems plausible because the thalamic mechanism underlying spindle generation involves reciprocal connections between excitatory and inhibitory cells, and activations of h and T currents[3,4], all of which are present in human supragranular cortex[53]. Furthermore, the consistently higher spindle frequency in the human thalamus compared to cortex is hard to explain if cortical spindles are all directly driven by the thalamus[6]. This thalamocortical frequency difference increases over the course of a spindle, as the spindle spreads across the cortex, and is correlated with the amount of such spread (Gonzalez et al., unpublished). Direct cortico-cortical spindle propagation may be necessary in humans, who have ~1400 cortical neurons for every thalamocortical cell (calculations based on previously reported cell counts[54,55]).

In summary, we show here that human cortical neurons have a strong increase in firing during spindles, both tonically and at particular spindle phases. This contributes to enhanced co-firing within 25 ms, a pre-condition for STDP, by neurons separated by ~0.4–4 mm. Spindles are focally organized within this micro-domain and co-firing is greatest at short distances when spindle coherence is high. Multiple patterns of wave propagation occur both within and between spindles, and are associated with distinct co-firing sequences. Therefore, spindles are associated with highly organized and dynamic spatiotemporal patterns of neuronal co-firing that may facilitate plasticity within local cortical networks.

## Methods

**Participants and data collection.** Four adult patients (Table 1) with focal, pharmaco-resistant epilepsy underwent 4–21 days of continuous electro-corticography and invasive EEG recordings for the localization of seizure foci prior to resection. The decision to implant and the duration of implantation were based entirely on clinical grounds. While undergoing clinical recording these patients also underwent intracranial microelectrode recordings with the Utah Array (Fig. 1a; Utah Array – © 2020 Blackrock Microsystems, LLC). The Utah Arrays were implanted after thorough review by the duly constituted Institutional Review Board (IRB) of Partners HealthCare, the parent institution of Massachusetts General Hospital and Brigham and Women's Hospital, which are academically part of Harvard Medical School. The IRB follows procedures defined by the US Department of Health and Human Services and is certified by that institution. In all cases, the Utah Array was implanted in a location that was strongly suspected based on pre-implant clinical information to be included within the boundary of the therapeutic resection, and in all cases it was later resected in order to gain surgical access to the focus which lay in deeper structures. The resected tissue in which the Utah Array was implanted was determined not to be an epileptogenic zone in any of these patients. No seizures originated from the area implanted with the Utah Array in any of the patients included in the study and no seizures occurred during the epochs analyzed. The arrays were implanted for research purposes and did not affect clinical monitoring. All ethical regulations for work with human participants were followed, and fully informed written consent for research was obtained prior to implantation of the Utah Arrays from all patients included in this study according to the Declaration of Helsinki guidelines as monitored by the local IRB at Partners HealthCare.

**Electrodes and localization.** The Utah Array is a $10 \times 10$ microelectrode grid, with corners omitted, that has 400 μm contact pitch (Fig. 1a). Each silicon probe is 1 or 1.5 mm long (summarized in Table 1) and 35–75 μm wide at its base, tapering to 3–5 μm at the tip, and is insulated except for the tip, which is platinum-coated.

Each patient in the study had one array implanted into the superior or middle temporal gyrus, in a region that was outside of the epileptogenic focus but which had to be removed in order to gain surgical access to the focus. Probes were placed under direct visualization perpendicular to the cortical surface. Based on a previous histological examination of human brain tissue, temporal cortex layer II begins at a mean of 252 μm and layer III ends at a mean of 1201 μm[56]. Therefore, we expect that the 1.0–1.5 mm long probes of the Utah array were implanted in supragranular layers II/III and possibly as low as upper granular layer IV.

**Recording and preprocessing**. Data were acquired at 30 kHz sampling (Blackrock Microsystems), from 0.3 to 7.5 kHz. Data were subsequently low-passed at 500 Hz and down-sampled to 1 kHz for the LFPs. Data were saved for offline analysis in MATLAB 2019b (MathWorks). LFPs were visualized in MATLAB: FieldTrip[57]. Channels were excluded when there were large amounts of noise or no units detected. Out of the 96 recording channels the mean number excluded from analysis was 29.75 (range 13–47). The 1 kHz data was average-referenced to negate the effects of the distant subdural reference, which could have detected neural activity distant from the array.

**Sleep staging and data selection**. After the data were collected, NREM sleep periods were determined during overnight periods by a neurologist trained in sleep staging according to the standard guidelines[58] as N2 (corresponding to S2 in the prior terminology) and N3 (combining S3 and S4 from the prior terminology), based on visual examination of 30 s epochs of electrocorticography data and concurrent video recording of the patient. All data analyzed were from overnight sleep. The night and sleep periods to be analyzed were chosen based on quality of sleep, quality of recordings, absence of ictal events, and time since prior ictal event. We required that there be at least 60 min of overnight N2/3 epochs that did not include seizures or large amounts of epileptic spiking as determined visually. Periods marked as N2 and N3, based on the presence of slow waves, K-complexes, and spindles, were selected for analysis. These periods were validated as N2 and N3 based on increases in delta (0.1–4 Hz) and sigma (10–16 Hz) band powers. Data from patients were only analyzed if the Utah Array was implanted in tissue that was determined clinically as not epileptogenic. Based on these criteria, four patients were selected for inclusion in this study out of ten implanted with Utah Arrays. One limitation to our study is that due to the clinical context (e.g., patients being awoken to measure vital signs), the sleep architecture of the patients could be disrupted.

**Spike detection and sorting**. The 30 kHz data recorded from each electrode contact was bandpassed at 300–3000 Hz with an 8th order elliptic filter with a pass-band ripple of 0.1 dB and a stop-band attenuation of 40 dB. Putative unit spikes were detected when the filtered signal exceeded 5 times the estimated standard deviation of the background noise[59], computed as:

$$\text{spike threshold} = 5 \times \frac{\text{median}(|x|)}{0.6745} \qquad (1)$$

Where $x$ is the 300–3000 Hz bandpassed data. The first three principal components of each spike were computed and unit clusters were manually selected. The remaining data points underwent clustering by k-means and a Kalman filter mixture model[60]. Visual inspection of the clusters identified by these two algorithms was used to determine which achieved better separation. Spikes were examined visually and those with abnormal waveform shapes or amplitudes far exceeding the majority of the spikes from their putative unit, such as those that may have been due to epileptiform activity, were excluded from analysis.

**Single unit classification**. PYs fire at low rates (~0.1 Hz in humans), with frequent bursting, and have short refractory periods and sharp spike autocorrelations, whereas INs typically fire at high rates (>1 Hz), with infrequent bursting, and have long refractory periods and broad spike autocorrelations. We classified units based on established methods in rodents[61,62] that have been adapted for use in humans[63], with additional considerations for MUs, which had spikes that were larger than the background noise and thus exceeded the detection threshold but could not be clustered into separable units. For each unit we computed the firing rate, valley-to-peak time interval, half peak width time interval, and bursting index (summarized in Table 1; Fig. 1c shows distinct clusters of PY and IN based on firing rate, valley-to-peak interval, and half width interval). As bursting results in a bimodal distribution of inter-spike intervals (ISIs), the bursting index was determined by running the Hartigan dip test for unimodality on the logarithm of distribution of ISIs[64]. Units were classified as putative PYs if they had spike rates of ~0.1–0.8 Hz, long valley-to-peak and half width intervals (Fig. 1d), sharp autocorrelations (Fig. 1e), and a bimodal ISI distribution (Fig. 1f) reflecting a propensity for bursting. By contrast, units were classified as putative INs if they had spike rates of ~1–5 Hz, short valley-to-peak and half width intervals, broad autocorrelations, and a predominantly unimodal ISI distribution (Fig. 1d–f). All single units were required to have a refractory period ≥1 ms. Units that had lower amplitude spikes and higher firing rates were classified as MUs (Supplementary Fig. 4a, b). While this overall classification method is indirect and INs, in particular, have heterogeneous spiking properties[65], previous studies using human extracellular

recordings have supported the classification of putative PYs and INs using similar metrics[66,67].

**Single unit quality and isolation**. In order to confirm that the PY and IN that we detected and classified were distinct single units, we evaluated the quality of the units according to criteria used by Kamiński et al[68]. The mean and standard deviation peak signal-to-noise ratio for PY was 8.82 ± 3.42 and for IN was 5.18 ± 2.85 (Supplementary Fig. 1a, b), indicating that the unit spikes well-exceeded the noise floor. Since the refractory period is ~3 ms, the percentage of ISIs <3 ms suggests the amount of single unit contamination by spikes from other units. The mean and standard deviation of the percent of ISIs <3 ms for PY was 0.28 ± 0.49% and for IN was 0.33 ± 0.58% (Supplementary Fig. 1c, d), which indicate very little single unit contamination by spikes from other units. Since multiple single units were detected on certain contacts, we evaluated the degree to which their single unit clusters were separable using the projection test[69], which measures the pairwise projection distance in units of standard deviations. The mean and standard deviation projection distance of PY pairs on the same contact was 94.01 ± 85.64 standard deviations and of IN pairs was 82.51 ± 83.24 standard deviations, indicating that when there were multiple units detected on the same contact they were highly separable (Supplementary Fig. 1e, f). To verify the temporal stability of units across the recordings, we divided each patient's recording period into quartiles and confirmed through visual inspection that the mean waveform shape and amplitude of each unit were consistent across quartiles.

**Sleep spindle detection and analysis**. Spindle detection was performed using a previously established method[26] that is primarily based on the standard criterion of sustained power in the spindle band (Fig. 1g). Each channel was bandpassed at 10–16 Hz using an 8th order zero-phase frequency domain filter with transition bands equal to 30% of the cutoff frequencies. Absolute values of the bandpassed data were smoothed via convolution with a tapered 300 ms Tukey window and median values of 10–16 Hz band amplitudes were then subtracted from each channel to account for differences between channels. The data were then normalized by the median absolute deviation. Spindles were detected when the peaks in the normalized data exceeded 1 for at least 300 ms, and onsets and offsets were marked when these amplitudes fell below 1 (see Supplementary Fig. 2a for durations). Putative spindles that coincided with large increases in lower (4–8 Hz) or higher (18–25 Hz) band power were rejected prior to analysis to exclude interictal epileptiform activity and broad spectrum artifacts, as well as 5–8 Hz theta bursts, which may extend into the lower end of the spindle range[7]. All data analyzed were visually inspected in an LFP viewer to ensure that there were no seizures or large artifacts (MATLAB: FieldTrip[57]). Subsets of spindles from each channel of each patient were also inspected visually to confirm that these events contained multiple prominent 10–16 Hz cycles and were not contaminated with interictal epileptiform discharges or artifacts. Spindle frequency was calculated by dividing the number of zero crossings in the spindle band by two times the spindle duration (frequency distribution shown in Supplementary Fig. 2b). Coherence between co-occurring spindles was computed by finding the magnitude squared covariance of the 10–16 Hz bandpassed spindle-spindle co-occurring epoch. Spindle propagation velocity was calculated as:

$$v = \frac{2\pi f d}{\varphi} \qquad (2)$$

Where $f$ is the spindle frequency, $d$ is the distance between the contacts recording the two waves, and $\varphi$ is the phase offset between the waves.

**Interictal spike rejection**. To reject events or baseline periods contaminated with interictal epileptiform discharges, we detected interictal spikes (IIS) when the z-score of the analytic amplitude of the 20 Hz highpass and the 200 Hz highpass of the signal exceeded 7. We excluded baseline periods, spindles, downstates, and upstates from analysis when they fell within ±100 ms of IIS. Since interictal epileptiform discharges may impact spindle generation[70], we confirmed that the spindles kept for analysis were not coupled to IIS (Supplementary Fig. 2c)

**Downstate and upstate detection and analysis**. To evaluate unit spiking during spindles that occurred in association with downstates or upstates, we detected downstates and upstates using a previously established method[6,7]. Data from each channel were bandpassed at 0.1–4 Hz and consecutive zero crossings within 0.25–3 s were detected. The top and bottom 20% of amplitude peaks between zero crossings were then selected. The average high gamma (70–190 Hz) analytic amplitude was found within ±100 ms of each peak. Since the polarity of downstates vs. upstates for each channel was not known, we determined whether the average peak-locked high gamma envelope was higher for positive vs. negative peaks for each channel and assigned the polarity of upstates (more high gamma) vs. downstates (less high gamma) accordingly. We then selected individual downstates and upstates for analysis only if the mean ±100 ms peak-locked high gamma envelope was for downstates less than the mean and for upstates greater than the mean of a −4000 to −2000 ms baseline period relative to the peaks. Therefore, this method confirms the channel polarity for downstates vs. upstates, and ensures that each event is associated with the expected change in high gamma. Spindles that

coincided with downstates were identified as those where the downstate peak preceded the spindle onset within 750 ms, and spindles that coincided with upstates were identified as those where the upstate peak followed the spindle onset within 500 ms (Supplementary Fig. 8). Isolated spindles, i.e., those that did not coincide with downstates or upstates, were identified as those that did not have a top 20% positive or negative amplitude peak, detected as described above, within ±1000 ms.

**Unit spike rate and phase analysis**. The percent of baseline spike rate during spindles for each unit detected on the same channel as the spindle was computed by multiplying 100 times the spike rate during all concatenated spindles on the unit's channel divided by the spike rate of all of the concatenated randomly selected non-spindle epochs that were matched in number and duration to the spindles and did not contain periods within ±100 ms of IIS. Spindle phases of unit spikes were determined by computing the Hilbert transform of the 10–16 Hz bandpassed signal and then finding the angle of the analytic signal at the times of the spikes. The circular mean spindle phase angle was computed for each unit using MATLAB: CircStat[71].

**Unit pair co-firing analysis**. Prior to analysis, co-occurring spindle periods were identified when there were co-occurring spindles on two channels. The first onset and last offset of the two spindles was used as the co-occurring spindle epoch. Non-spindle epochs were selected as the NREM periods when there was no spindle, with 100 ms padded before the onset and after the offset of every spindle, detected on either channel of a given unit pair. Unit pair co-firing of $PY_1$–$PY_2$, $IN_1$–$IN_2$, $PY_1$–$IN_2$, and $IN_1$–$PY_2$ during co-occurring spindles was compared to: (1) co-firing during non-spindle epochs, where 1000 sets of randomly selected epochs in between spindles (during baseline) that were matched in number and duration to the spindles, and (2) co-firing during "shuff-spindles" where spike times of each unit of the pair during the co-occurring spindles were randomly shuffled 1000 times. Unit pair co-firing was quantified by counting the number of spikes from one unit (e.g., $PY_1$) during the 25 ms preceding each spike from the second unit (e.g., $PY_2$). Pairs with units detected on the same channel were not included in the analysis. Unit pairs with significantly increased co-firing within 25 ms were tested for co-firing order preference. For each pair, the number of spikes during the 25 ms window before vs. after the spikes of the other unit were compared. Co-firing latency was compared to spindle phase lag by identifying unit pair co-firing within 25 ms during co-located spindles. The angular difference of spindle phases was used to compute the spindle phase lag for each co-firing event.

**Estimation of population of co-firing cells**. To estimate how many PY had greater co-firing during spindles with a given PY within a 4 mm radius for cells from layer III, beyond what would be expected from a simple increase in firing rate, we used cell counts from DeFelipe et al.[29], who showed that within a $50 \times 50\ \mu m$ column of anterior temporal cortex in humans there are ~35 cells in layer III. We multiplied by this by 0.8 to determine the approximate number of PY, 28, resulting in 11200 PY within a square millimeter column of layer III. We then used the following equation to estimate the number of PY that had greater co-firing with a given PY:

$$D \int_0^4 PA\,dr \quad (3)$$

Where $D$ is the estimated density of PY per square millimeter, $A = \pi r^2$, with radius $r$, and $P$ is the probability of significant co-firing at a given $r$:

$$P = P_{r=4} + \frac{(4-r)(P_{r=0} - P_{r=4})}{4} \quad (4)$$

Where $P_{r=0}$ and $P_{r=4}$ were approximated as 0.08 and 0.04, respectively, and P decreasing linearly with r, based on our results in Fig. 4a. Note that this estimate is conservative in that it only considers cells in layer III. DeFelipe et al.[29] estimated ~158 in all layers within the $50 \times 50\ \mu m$ column so the estimate would be scaled by 158/35 to account for all layers. Also, this estimate is conservative because it only considers PY, and it assumes that the co-firing continues to increase linearly from r = 0.4 to r = 0, whereas it may be supralinear in that region. Finally, this estimate is conservative because it only counts neurons with co-firing beyond that expected from the spindle-related firing rate increase, while P is >2.5x greater when including all co-firing relative to baseline.

**Spike time tiling coefficient**. The correlation of co-firing within ±25 ms between units was computed using the spike time tiling coefficient (STTC), which unlike the correlation index, is independent of firing rates[28]. For a given unit pair, for example, $PY_1$–$PY_2$, the STTC was computed as:

$$STTC = \frac{1}{2}\left( \frac{P_{PY_1} - T_{PY_2}}{1 - P_{PY_1}T_{PY_2}} + \frac{P_{PY_2} - T_{PY_1}}{1 - P_{PY_2}T_{PY_1}} \right) \quad (5)$$

Where $P$ is the proportion of spikes from a given unit in a pair that occur within ±25 ms of the spikes from the other unit in the pair, and $T$ is the proportion of the total recording time within ±25 ms of the spikes from a given unit.

**Spatial analysis of spindles and unit spiking**. The spatial layout of the recording array resulted in a highly variable number of contacts at different inter-contact distances. In order to approximately equalize the sample size in different distance bins, channel-pairs were grouped progressively at increasing inter-contact distances until a minimum number was attained, and then a new bin was begun. For example, for determining the spatial fall-off of spindle co-occurrence, for each channel pair the number of co-occurring spindles with any overlap was determined, and then binned with at least 100 minimum channel pairs per distance bin. If there were subsequent pairs after 100 that had the same inter-contact distance then the values were included within that same bin. The distance values plotted show the mean inter-contact distance for each bin. The same progressive binning method was used for the analysis of spindle coherences (minimum per bin = 100), unit spike rates as a function of time and spindle coherence (minimum per bin = 200–500), spindle phase lags (minimum per bin = 1000), unit spike rates (minimum per bin = 50), unit co-firing significances (minimum per bin = 50), and unit STTCs (minimum per bin = 30) as a function of inter-contact distances.

**Analysis of spindle propagation**. Visualization of sleep spindle propagation was done by z-scoring the 10–16 Hz bandpassed data from each channel, which reduces effects of the average reference, as well as finding the phase of the 10–16 Hz bandpassed data using the angle of the analytic signal. Prior to characterizing spatiotemporal patterns, epochs during which at least 20% of non-rejected channels were spindling were selected so that spatiotemporal analysis was limited to times when there was active spindling across the grid. Rejected channels were spatially interpolated by performing a 2D biharmonic spline interpolation (MATLAB: griddata) of the analytic signal and then extracting the real signal since the following analysis steps require no missing channels. Spindle spatiotemporal patterns of propagation were characterized for each spindle using the NeuroPatt Toolbox[30], which uses optical flow estimation and singular value decomposition to extract dominant spatiotemporal patterns from phase velocity vector field time series. This method is closely related to principal component analysis (PCA) as it reduces the dimensionality of the data to extract patterns that comprise the most variance. However, this method differs from PCA in that it extracts spatial modes that are vector fields, which represent spatiotemporal propagation patterns. The input data were bandpassed at 10–16 Hz and the analysis was performed according to the phase within this band for each channel. The smoothing and non-linearity penalty optical flow parameters were set to $\alpha = 0.5$ and $\beta = 1$, respectively, and real singular value decomposition was performed on the velocity vector fields for spatiotemporal mode extraction. To analyze a sufficient number of co-firing events in relation to propagation patterns, the same spatiotemporal analysis was performed but with concatenated spindles for each patient.

**Statistical analyses**. Unit spike rates during spindles vs. baseline (randomly selected non-spindle epochs during NREM that were matched in number and duration to the spindles) were evaluated by testing if the ratio minus one was significantly different than zero by using a one sample two-sided Wilcoxon signed-rank test. To test whether there were differences between spindle conditions (e.g., isolated spindles vs. spindles that coincided with upstates or downstates), the ratios were evaluated using a paired two-sided Wilcoxon signed-rank test. To test if there was a difference between the spindle phase distributions of PY and IN, a parametric Watson–Williams multi-sample test for equal circular means was used. To test for spindle phase preferences of unit spiking, a Hodges–Ajne test was first used to determine if the distribution of the spindle phases of each unit was non-uniform. Next, the spikes were randomly shuffled 1000 times and the Hodges–Ajne test was used to determine the 1000 p-values of the spindle phase distributions. Finally, a unit was determined to have a significant phase preference if the p-value of its spindle phase distribution was in the 5th percentile of the 1000 p-values from the shuffled distribution. To compare unit pair co-firing during co-occurring spindles vs. baseline, we randomly selected 1000 sets of non-spindle epochs, during which no spindles in either channel were detected, and which were matched in number and duration to the spindle co-occurrence epochs. The p-value was calculated as the percent of the 1000 sets of randomly shuffled non-spindle epochs that had more spikes from one unit in the 25 ms preceding the spikes of the second unit. For example, for a given $PY_1$–$IN_2$ the number of $PY_1$ unit spikes within 25 ms preceding all $IN_2$ spikes was counted. To compare unit pair co-firing during spindles vs. shuff-spindles, where the times of the spikes of each unit during each co-occurring spindle were randomly shuffled 1000 times, the p-value was calculated as described above. To test whether there was greater co-firing for spindles that coincided with downstates or upstates (which were pooled due to the small number of events) vs. isolated spindles, we computed the average co-firing rate across unit pairs based on the spikes of $PY_1$ (or $IN_1$) within the 25 ms preceding the spikes of $PY_2$ (or $IN_2$) using a paired two-sided Wilcoxon signed-rank test. This same test was also used to confirm that there was more co-firing during isolated spindles vs. non-spindles. The significance of ordered co-firing of each unit pair was computed by comparing the proportion of the spikes from one unit occurring in the 25 ms before vs. after the spikes from the other unit using a two-sided $\chi^2$ test of proportions (MATLAB: prop_test[72]) for all pairs with a minimum of 10 co-firing events during the ±25 ms windows. To evaluate spindle co-occurrence vs. chance, we compared the spindle co-occurrence density vs. the chance spindle co-occurrence density using a paired two-sided t-test. For each

channel pair the chance spindle co-occurrence density was determined by randomly shuffling each channel's inter-spindle intervals 100 times and finding the mean density of chance co-occurrences. Significance was evaluated for $\alpha = 0.05$, and when indicated a Bonferroni-corrected $\alpha$ was used to account for multiple comparisons. All fits were approximated with a linear least squares regression, and for fits with $R^2 < 0.3$, exponential least squares regressions were instead used if they met $R^2 > 0.3$. If both fits met $R^2 > 0.3$ then a linear fit was used unless the $R^2$ of the exponential fit was >10% larger. Fits are only shown for significant linear relationships or well-approximated exponential relationships. To test for the significance of a linear relationship, the significance of the correlation coefficient was used. To visualize the instantaneous phase of a propagating spindle, a cyclic color map was used[73]. To generate controls for the analysis of spatiotemporal propagation patterns, we randomly shuffled the positions of the good channels once for each spindle prior to interpolation. The Cohen's d was calculated according to MATLAB: computeCohen_d[74].

**Reporting summary**. Further information on research design is available in the Nature Research Reporting Summary linked to this article.

## Data availability
The authors' data sharing agreement currently does not permit making the raw data publicly available. The raw data that support the findings of this study are available from the corresponding authors upon reasonable request. Source data are provided with this paper.

## Code availability
The code that support the findings of this study are available from the corresponding authors upon reasonable request.

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

## Acknowledgements

Supported by NIMH (RF1 MH117155, T32 MH020002), ONR-MURI (N00014-16-1-2829), NIBIB (R01 EB009282), and the Kavli Institute for Brain and Mind. We thank Adam Niese, Burke Rosen, Christopher Gonzalez, Daniel Cleary, Erik Kaestner, Jacob Garrett, Sophie Kajfez, Xi Jiang, Yihan Zi, and Zarek Siegel for their support.

## Author contributions

C.D. and E.H. designed the study. S.C., E.E. and J.M. collected the data. C.D. and A.S. performed the analyses. C.D. and E.H. wrote the manuscript. E.H. and S.C. supervised the work.

## Competing interests

The authors declare no competing interests.
