## [Peer Review File · Nature Communications]

REVIEWER COMMENTS

Reviewer #1 (Remarks to the Author):

Overall, the authors have very carefully and satisfactorily addressed all of my comments. Two minor points remain:

1. The authors determined sleep stages based on standard criteria by Rechtschaffen and Kales (1968). Based on these criteria, NREM sleep is divided into S1 to S4. Spindles are typical for S2-S4. However, it appears that the authors have now based their analyses only on S2 and S3. It is unclear if and why S4 has not been included. Please, clarify and make sure that S4 is included.

2. In the rebuttal letter, the authors indicate that "all patients who had a Utah Array implanted into tissue that was determined to be epileptogenic were excluded from the study". However, it is unclear if and how many patients had to be excluded based on this criterion. This information needs to be added to the ms.

Reviewer #2 (Remarks to the Author):

The authors have performed extensive additional analysis, but they have not addressed several issues adequately. I am elaborating on each point below, with the authors' rebuttal text. Overall, it is not clear from the presented results if the co-firing during spindles is inducing neural plasticity or that it has relevance for memory consolidation. First, plasticity is by definition accompanied by the connectivity weight changes. The authors did not attempt to demonstrate the change in co-firing statistics as a function of co-firing history, which would resemble the plasticity-induced weight changes. Second, there is no evidence that the described neuronal dynamics during spindles are selective for neuronal pairs involved in memory consolidation, as no memory task was performed. With these deficiencies and the methodological problems listed below, the manuscript, in my opinion, is not suitable for publication in the current state.

1) Determination of chance level.

(6% of all pairs significant with 0.1% expected by chance for $\alpha=0.001$)

(with 5% expected by chance for $\alpha=0.05$)

What is the basis for calculating chance levels and why is it different in these two cases?

2) Unit stability criteria.

We also would like to note that none of our results would be expected to be caused by unit temporal instability.

For the recordings lasting for several hours, neuronal drift is quite common, and it could affect the results presented in the manuscript. For example, suppose the two neurons decrease firing in parallel due to neuronal drift. In that case, the co-firing statistics could be artificially inflated due to the fact that BOTH neurons don't fire during the later spindle episodes. Therefore, the authors need to use the stability criteria to either exclude the units with unstable firing from the analysis or limit the analysis of a given pair to the period when both units were showing stable firing.

3) Spindle detection criteria. The authors justified using a very short lower spindle duration threshold by claiming that intracranially recorded spindles are very short duration events and offer the previous paper from their lab as the evidence for acceptance of these criteria (Hagler et al., 2018). However, they are probably well aware that the most cited literature on intracranially recorded spindles in humans was based on 500 msec lower threshold (Nir et al., 2011; Andrillon et al., 2011), as well as the vast literature from rodents. The histogram of spindle durations indicates that most detected events were shorter than the standard in the field. Since the authors claim that there is no difference between the spindles shorter and longer than 500 msec, they should demonstrate their results using the broadly accepted spindle duration definitions (>500 msec).)?

3) Connectivity determination.

"Coherence is sensitive to the covariance, or joint variance, or how much the amplitude of two signals changes together, although it is not sensitive to the amplitudes of the individual signals since in computing the coherence the variance of both signals is divided out. We propose that a purely phase-based measure, such as phase-locking value, has less biological relevance since co-amplitude is relevant to the underlying neuronal activity. Two small signals can have significant phase-locking, but this is not necessarily biologically meaningful.

It has been demonstrated that the signal amplitude can affect the coherence estimate (Srinath and Ray, *J of Neurophys*, 2014):

"Importantly, even when phases were completely randomized between two electrodes, amplitude correlations introduced significant coherence. To quantify the contributions of phase consistency and amplitude correlations to coherence, we simulated pairs of sinusoids with varying phase consistency and amplitude correlations. These simulations confirmed that amplitude correlations can significantly bias coherence measurements, resulting in either over- or underestimation of true phase coherence."

In addition, if the authors are making a case that the directional co-firing of a neuron pair is essential for plasticity induction and neurons are phase-locked to local spindles, it would be expected that the spindles on the electrodes where those neurons are recorded show consistent phase relation; hence, the most relevant connectivity measure would be phase-based. If the authors are concerned that low-amplitude events would contaminate phase-based connectivity estimates, they could simply use the appropriate amplitude threshold. Using the standard duration threshold for detecting spindle events (500 msec, as mentioned above), would also help avoid low amplitude events.

Reviewer #3 (Remarks to the Author):

The authors address all comments in great detail. I also found their responses to the other reviewers comments valuable. I have no additional comments. The manuscript likely constitutes a valuable resource for future studies. Therefore, it'd be desirable if at least some of the data could be made available to replicate how firing patterns change relative to field potentials. In terms of interpretation, it should be made sufficiently clear that all data stems of anterior lateral temporal cortex, which is not part of the network that is typically associated with sleep-dependent memory consolidation (since the authors rest the hypothesis on theoretical work as reviewed in e.g. Klinzing 2019, which very much focuses on the mPFC-MTL pathway).

Responses to Reviewers
9/2/20

We thank the reviewers for their excellent and thoughtful feedback on our manuscript. The major concerns from the reviewers included that the methods were not sufficiently described, unit quality and isolation was not clearly demonstrated, detection and exclusion of epileptic activity was not adequately performed, spindle selection was not fully validated, and that our observations were not clearly linked to mechanisms of plasticity. We have substantially revised our manuscript and the supplementary information, which in addition to updating the text involved running new analyses as well as re-running nearly all of the existing analyses, in order to address these and the other concerns raised by the reviewers. As a result, we believe that our manuscript is much stronger and worthy of further consideration.

Reviewer #1 (Remarks to the Author):

The authors investigated the role of traveling sleep spindles for spike-timing-dependent plasticity (STDP) using electrocorticography and intracranial microelectrode recordings implanted into the superior or middle temporal gyrus of four patients with focal, pharmaco-resistant epilepsy. They found during sleep spindles an increased firing within and between cortical units (presumed pyramidal cells and/or interneurons, greatest within ~2 mm distance) during a time interval of 25ms, necessary for STDP (as previously suggested in rodents). Spindles propagated in specific patterns at a velocity of ~0.23 m/s within a sub-centimeter scale, possibly facilitating plasticity across multiple networks during non-rapid eye movement (NREM) sleep.

The findings are highly interesting for the sleep and memory community, and the manuscript is well-written. However, data analyses appear to be not always described in sufficient detail. Please find my concerns listed below.

Major issues

- The criteria used for electrocorticography-based sleep staging are unclear. Please describe in more detail and/or cite respective literature. Does this electrocorticography-based sleep staging correspond to rules applied to EEG-based sleep staging (i.e., by Rechtschaffen & Kales or AASM)? How long were the "periods"/epochs used to classify sleep stages? How were other sleep stages such as stage 1 sleep or REM sleep classified? Please also provide a table indicating standard sleep architecture for each participant (i.e., total sleep time, time/percentage spent in stage 1, 2, 3 and REM sleep, wake after sleep onset, sleep efficiency, number of spindles, spindle density, number of epochs with potential epileptic seizures etc.).

We updated the Methods ("Sleep Staging and Data Selection") to explain that after the data were collected, NREM sleep stages 2 and 3 were determined by a neurologist trained in sleep staging following the standard guidelines promulgated by Rechtschaffen and Kales (1968), and based on visual examination of successive 30s epochs of electrocorticography data and concurrent video recording of the patient. REM and waking were only tentatively identified because EMG and EOG were not available in most patients. Thus we cannot provide a standard sleep architecture table. The slow waves and sleep spindles which are characteristic of NREM stages 2 and 3 are clearly visible in the ECoG data, and we extracted those stages from the data to perform all of our analyses of the carefully selected spindles themselves as well as the periods in between them. Thus, we would suggest that additional information on sleep architecture is not required because it would not affect the conclusions of our study. In the paper, we report spindle metrics, including numbers and density, and show confirmation that no epochs with epileptic seizures were included in the data analyzed. Furthermore, we note as a limitation of our study that due to the clinical context (e.g., awakening to measure vital signs), the sleep architecture of the patients could be disrupted.

- There is evidence that sleep spindles are different in patients with epilepsy, and this further depends on the type of epilepsy (e.g., Drake et al., 1991, Clin Electroencephalogr). Moreover, spindles may contribute to triggering nocturnal seizures (Montplaisir et al., 1981, Sleep). How can the authors rule out that the

results are representing a general mechanism (that applies to healthy participants) that is not an atypical phenomenon due to epilepsy – especially, when the location of recording was strongly suspected and later resected? Did participants have epileptic seizures during recordings? How were interictal spikes identified and how was their occurrence related to the occurrence of spindles? etc.

We updated the text to make it more clear that in the rare cases where the Utah Array could be implanted, the location of the array was in tissue that was very strongly suspected of being within the ultimate resection zone, and that was always the case. However, the resection zone includes both the tissue that is the location of the epileptic focus, and the surrounding tissue that has to be excised to provide access to the focus. For example, in the standard anterior temporal lobectomy, the most anterior portion of the lateral temporal neocortex is removed to provide access to the underlying medial temporal lobe, even though the anterolateral temporal cortex is found not to be engaged at seizure onset. Such was the case in the patients included in this study. Conversely, all patients who had a Utah Array implanted into tissue that was determined to be epileptogenic were excluded from the study. No seizures originated from the area implanted with the Utah Array in any of the patients included in this study and no seizures occurred in the brain during the epochs analyzed. Furthermore, the background activity recorded by the Utah array appeared grossly normal, lacking the frequent spikes and waking slow activity often recorded in the epileptic focus.

Since it is not currently possible to record directly from the cortex of healthy subjects, it was necessary to use rare opportunities such as these recordings from epilepsy patients undergoing invasive monitoring in order to study the microphysiology of sleep spindles in human cortex. Drake et al. (1991) studied patients with different types of epilepsy and found that all patients had 12-13 and 14-15 Hz spindles, which are present healthy subjects (Fernandez and Lüthi, 2020). We found an average spindle frequency of 12.52 Hz, and 56% of the spindles we detected and analyzed were within 12-14 Hz (see added Supplementary Fig.2B), which is consistent with the definition of the American Association of Sleep Medicine that spindles are most commonly 12-14 Hz (Iber C, 2007). The distribution of spindle frequencies came from a standard normal distribution ($p \approx 0$, one-sample Kolmogorov-Smirnov test). Drake et al. (1991) found that ~10 Hz spindles varied by type of epilepsy, suggesting that they may not be representative of spindles in healthy subjects, but in our data only 8.5% of spindles were <11 Hz. Notably, they suggested that rhythmic activity in the alpha frequency range may be associated with pathology, but our spindle detection method rejected events with increased activity below (and above) the spindle frequency range of 10-16 Hz, to minimize the detection of non-spindle events, including alpha, whose frequencies may extend into the spindle range.

In this revised manuscript, we detected inter-ictal spikes (IIS) on each channel when the z-score of the analytic amplitude of the 200Hz highpass and the 20Hz highpass exceeded 7. We confirmed visually that this procedure reliably excluded events that were potentially epileptiform in the data. As we note, they were rare, as would be expected given the selection criteria above. We removed spindles and non-spindle epochs within +/-100ms of these events and re-ran all analyses. In added Supplementary Fig.2C we report a peri-IIS time histogram of spindle onsets to demonstrate that spindles associated with IIS were effectively removed and that there were no delayed associations between IIS and spindle onsets.

In sum, we have now carefully removed spindles and non-spindle epochs that were potentially associated with IIS and the spindles included in our analyses oscillate specifically within the spindle band and at frequencies consistent with those found in healthy subjects. While it would be preferable to study these phenomena in healthy subjects, we feel that we have taken sufficient measures to include only spindles that are most likely to be physiological in the only population of humans (i.e. neurosurgical patients) from whom direct recordings from the brain are currently possible.

- I am not convinced that the results on phase-locked unit spiking during spindles are robust, as they seem to be based on roughly 30 pyramidal cells and 20 interneurons only.

At $\alpha=0.05$, we expected that 5% of PY and 5% of IN would have significant spindle phase locking, however we found that more than 3x this amount of PY and 12x of IN were phase-locked.

In this revision, we ran an additional analysis suggested by Reviewer #3 to compute the mean spike rates of PY and IN time-locked to spindle trough (see added Supplementary Fig.7). We found a strong locking of PY and IN spiking to the spindle trough, which supports our conclusion that unit spiking locks to spindle phase.

- Given the small number of patients, different ages, different probes, different location of probes, different hemispheres and gender differences, the authors should indicate in more detail if results were comparable between these parameters. For instance, in the results section (“Spindles are associated with an increase in unit spike rates”), the author indicated that there are similarities as well as differences for 1.0 mm and 1.5 mm probes, but they do not show any data or statistics (on the group level). It is further unclear why the authors used different silicon probes of different length and width for the four patients in the first place. For how long (hours/nights) were intracranial recordings performed in each patient? Were recording periods of similar duration for the four patients? How was the data averaged across the four patients in case of different recording lengths?

We added patient specific recording durations to Table 1. Furthermore, we added detailed results for individual patients for key statistical tests in the paper (Supplementary Tables 1 and 2), which should help to alleviate the concern that we did not explicitly take into account different recording lengths when averaging across patients. We found generally consistent results across patients. Due to the small number of patients we were not able to test for differences based on characteristics such as age and sex, but we note that our spindle metrics such as frequency and density are within the expected ranges for healthy subjects across all four patients. Data availability was limited to recordings from arrays with different probe lengths, and we have made sure that the interpretations of our results consider the difference in the estimated laminar locations of the contacts based on these different probe lengths. The array contacts from which the data were recorded had the same approximate widths (3-5 μ m) across contacts and patients. These are rare patients and thus their number is limited. We also updated our quantification of unit spiking to replace the original baseline period, which was all of the epochs in between spindles, with a baseline period that consisted of randomly selected epochs matched in number and duration to the spindles. While the Utah Array was implanted in either superior or middle temporal gyrus, previous work has shown that spindles in these two locations are very similar (Piantoni et al., 2016).

- How can the authors exclude that the signal of one presumed propagating spindle detected at different channels/locations does not actually correspond to two different spindles?

The characteristics of the spindles with overlapping time courses in different channels indicate that they are related but have distinct generators. We show in Fig.5-6 and Video 1 that spindles have organized patterns that propagate to adjacent channels. We found high coherence between co-spindling channels and a strong linear relationship between spindle phase lag and distance. The high coherence shows that the spindles recorded with overlapping time courses by different channels are not independent. The non-zero phase lag shows that the spindles recorded by different channels have different generators. The linear relationship between phase and distance shows that the spindles are spatially organized across the array. We also found a significantly greater percent explained variance of the dominant spatiotemporal mode of spindles for actual vs. shuffled channel positions, further indicating that the propagation pattern was organized across the grid, which is consistent with unified spindling vs. many independent, uncoordinated sources. While we have not ruled out the possibility that there are instances of independent spindle patterns propagating across the array simultaneously, this would still be consistent with the conclusions in this paper, including that spindles have multiple propagation patterns that entrain different co-firing sequences.

- Referencing needs to be updated. Mechanisms and functions of spindles have recently been comprehensively reviewed by Fernandez and Lüthi A (Physiol Rev 2020), and a review of spindles for memory consolidation has been provided by Klinzing J, et al. (Nat Neurosci 2019). I also miss ground laying early work by Timofeev/Steriade who were the first to propose a plasticity function (“augmenting response”) of spindles as well as recent work (e.g. Niethard N et al, PNAS 2018) supporting this view. All this work should be integrated to provide a more balanced picture.

All of these references have been added to this revised manuscript.

Minor issues

- p.15, last sentence: I do not follow the logic of this calculation. If 17% and 11% of neurons exhibiting phase preference were pyramidal cells and interneurons, respectively, what are the remaining 72%?

We removed this sentence since it was confusing and did not add information nor support further understanding.

- Firing and co-firing during spindles might be generally increased during slow-oscillation upstates in slow-wave sleep. Did the authors also have the chance to look at spindle/slow oscillation coupling?

To address this, we performed further analyses by detecting downstates and upstates according to Gonzalez et al. (2018) and Mak-McCully et al. (2017), where data from each channel were bandpassed at 0.1-4 Hz and consecutive zero crossings within 0.25-3 s were detected. The top and bottom 20% of amplitude peaks between zero crossings were then selected. The average high gamma (70-190 Hz) envelope, computed as the average amplitude of the analytic signal, was found within ± 100 ms of each peak. Since the polarity of downstates vs. upstates for each channel was not known, we determined whether the average peak-locked high gamma envelope was higher for positive vs. negative peaks for each channel and assigned the polarity of upstates (more high gamma) vs. downstates (less high gamma) accordingly. We then selected individual downstates and upstates for analysis only if the mean ± 100 ms peak-locked high gamma envelope was for downstates less than the mean and for upstates greater than the mean of a -4000 to -2000ms baseline period relative to the peaks. Therefore, our method confirms channel polarities for downstates vs. upstates, and ensures that each event is associated with its expected change in high gamma. See added Supplementary Fig.3A-B for the mean LFP and high gamma envelope of downstates, added Supplementary Fig.3D-E for the upstates of a representative channel.

In added Supplementary Fig.3C,F we show an example peri-spindle onset time histogram of downstates (C) and upstates (F). We found that spindle onsets tended to occur within ~ 750 ms following downstate peaks and within ~ 500 ms preceding upstate peaks, which is consistent with the literature. In this revision, we report PY and IN spiking for all spindles (as we reported in our initial submission), spindles coinciding with downstates (spindle onsets occurring within 750ms following downstate peaks) or upstates (spindle onsets occurring within 500ms before upstate peaks), and isolated spindles that do not have positive or negative peaks (i.e. upstates and downstates without high gamma validation so as to be highly selective) within ± 1000 ms. Please note that we pooled spindles coinciding with downstates or upstates since these events tended to happen in conjunction with spindles tending to occur on the down-to-upstate transition (see added Supplementary Fig.3 and 9), there were relatively few events, and both conditions were associated with increased unit spike rates, but these increases were not significantly different from each other (please see section “Units increase spiking more during spindles that coincide with down-upstates” in the Results as well as added Supplementary Fig.9 for separate results for spindles coinciding with downstates vs. upstates).

We also found that isolated spindles (those that did not coincide with downstates or upstates) were associated with increased unit spiking compared to baseline (Fig.2D,F). Furthermore, co-firing was increased during isolated spindles vs. baseline and further increased during spindles that coincided with downstates or upstates vs. isolated spindles (Fig.3A-H, added Supplementary Fig.10). Together, these results show that unit spiking increases during spindles (even when they

do not coincide with downstates or upstates) and that this increase is enhanced when spindles coincide with downstates or upstates, which may therefore help to explain why spindles that occur on the down-to-upstate transition are especially important for memory consolidation. Please note that we did not do further analyses due to the relatively small numbers of spindles that coincided with down-upstates compared to the large number of spindles necessary for many of our analyses.

- In the Fig.1 caption for panel G, the authors state that they are showing raster plots of unit spiking of the “three” putative cell types, while in fact they are only showing raster plots for two cell types (PY, IN). Please adapt.

We made this change.

- The definition of “shuff-spindles” (sleep spindles with shuffled spikes) should be introduced at first mention in the main text.

We added the definition there.

- Typo in Discussion, third paragraph: should read “STDP”.

We fixed this typo.

Reviewer #2 (Remarks to the Author):

Dickey et al. analyzed the spatiotemporal patterns of putative spindle activity during sleep (NREM2-3) and associated single unit firing recorded from the human brain using Utah arrays. They explore a valuable dataset with a large number of simultaneously recorded single units. The main claim of the paper is that spindles synchronize single neurons at short cortical distances, enabling the plasticity changes. This finding could contribute to our understanding of brain plasticity during sleep, possibly related to memory consolidation. However, the proportion of pyramidal cell pairs with ordered co-firing during spindles seems to be <2% (Table 2), which raises the possibility that it might happen by chance. In addition, several methodological issues require careful clarification before the results could be properly interpreted and conclusions evaluated.

We apologize that our presentation of the data was confusing resulting in the impression that only 2% of the PY cell-pairs show changes which could support STDP. Perhaps this was an unintended result of our effort to parse the various conditions which could work separately and/or in concert to support STDP. For PY pairs, we found that co-firing during spindles vs. non-spindles far exceeded chance (16% significant with 0.1% expected by chance for $\alpha=0.001$). Statistically, an increase in co-firing could be due to simply increased firing. In order to assess this possibility, we compared co-firing for cell-pairs during spindles with the actual spike times vs. shuffled spike times, and found that many remained significant (6% of all pairs significant with 0.1% expected by chance for $\alpha=0.001$). Thus, 16% of PY-pairs increased co-firing during spindles, and in 38% of these, the increased co-firing significantly exceeded that which would be expected given only the increase in firing rate. Considering now only the PY pairs that satisfy both of these conditions, 19% showed significant directional specificity (with 5% expected by chance for $\alpha=0.05$). For ordered spiking, we selected only to analyze units with significantly increased co-firing for both of the co-firing tests since these are the units to be expected to participate in plasticity. We have added a brief analysis addressing whether this level of co-firing is functionally meaningful. Taking the data for human anterior MTG and STG from DeFelipe 2002, and assuming co-firing is confined to layer 3, we estimated that ~37,500 PY within a 4mm radius would increase co-firing during spindles with a given PY beyond that expected from a simple increase in firing rate.

In the revised manuscript, we address the methodological issues described below so that the results can be properly interpreted and conclusions evaluated.

1) Data inclusion: The recordings were performed continuously for 4-21 days/subject, and one dataset/subject was selected for the analysis. Was the selection based on pre-established objective criteria, or was it random? Were any datasets excluded at this stage? It is unclear if the data was collected during overnight sleep, sleep periods during the day, or both. If the data was collected during overnight sleep, did the authors include in the analysis all the NREM2-3 periods recorded during the same night? If not, what was the basis for selection?

We updated the methods to make the selection criteria explicit. First, we selected patients in whom the Utah Array was implanted in tissue overlying the epileptic focus, which had to be removed to surgically access the focus, but that was determined not to be the site of ictal onset itself, and which lacked frequent epileptiform discharges or visibly abnormal background activity. All data analyzed were from overnight sleep. The night and sleep periods to be analyzed were chosen based on quality of sleep, quality of recordings, absence of ictal events, and time since prior ictal event. We required that there be at least 60 minutes of overnight NREM2/3 epochs that did not include seizures or large amounts epileptic spiking determined by visual inspection.

2) Single unit isolation quality and stability: It is unclear what isolation quality criteria were used to select single units. The only hint is given on page 7, noting that single units were required to have refractory periods of ' ≥ 1 ms'. The refractory period is 3ms and the percentage of spikes during this period is an indication of single unit contamination by spikes from other units, which is a criterion for unit selection. The authors should show detailed unit quality metrics and use the inclusion criteria widely accepted in human single unit literature (e.g. Kaminski et al., 2020; Supplementary Fig.2). Finally, for recordings lasting several hours or overnight, neuronal drift might affect the results, so the authors should use temporal stability as the criteria for unit inclusion.

We performed several additional analyses according to Kaminski et al. (2020) to show that the units we selected have large signal-to-noise, are well isolated, and are temporally stable. Please see the "Single Unit Quality and Isolation" section added to the revised manuscript for details. We also would like to note that none of our results would be expected to be caused by unit temporal instability.

3) Removal of epileptic activity: The authors stated that all the detected spindles were visually inspected for the absence of epileptic activity (Page 8). However, the number of detected spindles is ~500000 (Page 12), which doesn't seem feasible for human visual inspection. Even if the spindles were visually inspected, what about the non-spindle periods used for baseline?

We intended to convey that many but not all spindles from each channel from each patient were reviewed visually to verify that the automatic spike detection was effective in rejecting all spikes. We updated the Methods to make this clear. All of the data, including the non-spindle periods were visually inspected in an LFP viewer to ensure that there were no seizures. To further ensure that spindles and non-spindle epochs did not contain epileptic activity, we repeated inter-ictal spike detection with lower thresholds and re-ran our analyses eliminating not only the period contaminated but the preceding and following periods. Please see the "Interictal Spike Rejection" section in the revised manuscript for details.

4) Spindle detection: Spindles are defined across the rodent and primate literature as lasting 0.5-2/3 sec, and the authors note this in the introduction (Page 2). However, criteria for spindle detection used in this paper include the events as short as 200ms, with the mean duration ~400ms. This suggests that the large proportion of analyzed events are as short as 2-3 oscillatory cycles, which raises the possibility of false positives. For these results to be comparable with other literature (e.g. Nir et al., 2011; Andrillon et al., 2011), the authors should use the widely accepted spindle detection criteria.

We mistakenly reported that the minimum spindle length was 200ms, which is the original minimum duration parameter used in the Hagler et al. (2018) method, when we had in fact raised this minimum to 300ms. We updated the text accordingly and confirmed that the rest of the

parameters reported are accurate. Also in this revision, we added Supplementary Fig.2A to show the distribution of spindle durations.

Spindles recorded directly from the brain may have shorter durations than those recorded at the scalp because scalp recordings have low spatial resolution and may detect consecutively activated spindle generators. Indeed, previously published intracranial studies have included spindles with durations shorter than 0.5s. Hagler et al. (2018) analyzed intracranial laminar microelectrode recordings in humans and included spindles down to 200ms and Srihari (2020) analyzed intracranial microelectrode recordings in macaques and found spontaneous sleep spindle durations down to ~300ms (extending up to only ~1.2s).

To confirm that the spindles we included that had durations shorter than 500ms (i.e. shorter than the standard lower limit included in studies using scalp recordings) were similar to those with durations longer than 500ms, we analyzed unit spiking during spindles with these two sets of durations. There was a significant increase in unit spiking during spindles compared to baseline for both shorter and longer spindles, but there was no significant difference between these increases for short vs. long spindles (added Supplementary Fig.4), which supports the decision to include spindles shorter than 500ms in our analyses.

5) Rigor and reproducibility: The authors reported a large increase in single unit firing during spindles, contrary to Andrillon et al. (2011). The baseline for this comparison is defined as periods outside of the spindle epochs, likely include cortical downstates, a period during which most of the neurons are silent by definition (see the raster plots in Fig.1G). To demonstrate the firing rate increases during spindles, downstate periods should be detected based on accepted criteria and excluded from the baseline. This detail is also critical for other comparisons that use non-spindle baselines.

Rather than exclude the many events that occur between spindles and may be associated with changes in firing, including downstates, upstates, and theta bursts, we considered these events as part of the non-spindle baseline. Of note, Andrillon et al. (2011) measured unit spiking before, during, and after 1) spindles following downstates and 2) all spindles (please see Fig.7A in Andrillon et al., 2011).

In this revised manuscript, we generated spindle onset-locked unit spiking plots in the format used by Andrillon et al. (2011). Our results clearly demonstrate an increase in unit spiking during spindles compared to ± 1500 ms around the spindle (Fig.2A). Furthermore, we detected downstates and upstates and identified spindles that coincided with downstates (where spindle onset followed downstate peak within 750ms), spindles that coincided with upstates (where spindle onset preceded upstate peak within 500ms), or isolated spindles (those that did not occur within ± 1000 ms to downstates or upstates). Our detection method and further details are provided in the updated Methods and added Supplementary Fig.3. We also generated the spindle onset-locked unit spiking plots for spindles that coincided with down-upstates (i.e. downstates and/or upstates; Fig.2E), downstates (added Supplementary Fig.9A), upstates (Supplementary Fig.9C), or isolated spindles (Fig.2D). In all cases there was a clear and statistically significant increase in spiking during spindles. We believe that these new analyses provide sufficient confirmation that spiking increases during spindles and that this is not due simply to downstates in the baseline.

Furthermore, we updated our quantification of unit spiking such that instead of using a baseline that consisted of all the epochs in between spindles, we used a baseline period that consisted of randomly selected epochs matched in number and duration to the spindles.

6) Spindle coherence: Coherence is a connectivity measure well known to be inflated by the oscillation amplitude. To account for this, the authors should either use the power-balanced design or a connectivity measure less influenced by amplitude (e.g. phase-locking value).

Coherence is sensitive to the covariance, or joint variance, or how much the amplitude of two signals changes together, although it is not sensitive to the amplitudes of the individual signals

since in computing the coherence the variance of both signals is divided out. We propose that a purely phase-based measure, such as phase-locking value, has less biological relevance since co-amplitude is relevant to the underlying neuronal activity. Two small signals can have significant phase-locking, but this is not necessarily biologically meaningful. We have updated the text to better explain why coherence was used in our analyses.

7) When describing the correction for multiple comparisons, the authors should define the basis for corrections in each case and provide the exact n.

We added this information in the Results for each case.

8) Spike time tiling procedure is not described in detail.

We added a more detailed description of this procedure and included its equation in the Methods.

9) When shuffling procedures are used to generate null distributions, the process should be precisely described (what is being shuffled? what shuffling method is used? within what time range?).

We updated the Results in each case when shuffling was mentioned to cover these three points.

Reviewer #3 (Remarks to the Author):

Review of Dickey et al.

In the present manuscript, the authors report data from 4 epilepsy patients who were undergoing presurgical evaluation with implanted electrodes. Here additional Utah arrays were implanted during invasive Phase-II monitoring. The authors focused on NREM sleep and report both field potentials as well as spiking data from putative single neurons. The work builds on a series of recent papers by the same group and is well-grounded in the literature.

Overall, the authors report a series of analyses that focus either on (a) LFP-spike locking, (b) spike correlations, (c) spatial correlations and trajectories of traveling waves. There are no pre-post sleep comparisons, no behavioral task and no quantification if any plasticity processes occurred. While the paper reflects an impressive tour-de-force using a variety of methods, the main conclusion that traveling spindle waves create the necessary conditions for long-term plasticity is not supported by the present data.

The key issue is of the present manuscript is that multiple analyses are being introduced taking advantage of a unique dataset, which is very difficult to obtain, but it lacks theory and a clear motivation. Some of the analyses appear rather arbitrary and only loosely connected. In the end, the reader is left with an observation of how firing changes during spindles, without further quantification. Then the spatial dimension is introduced, several correlations are reported, then the traveling waves idea is introduced and in the end the results are interpreted as reflecting a neurophysiological scenario that mediates STDP. The authors might want to highlight the observation, which supports the plasticity aspect.

We thank the reviewer for this comment and we have extensively revised the narrative so as to make clear the relation of the individual analyses to our overarching goal. Specifically, we seek here to arrive at an understanding how spindles provide a spatiotemporal context for plasticity which could underlie their previously demonstrated role in behavioral consolidation. We believe we have made it much more clear how changes in unit spiking – both tonic and phasic – contribute to increased co-firing within 25ms during spindles, as well as how spindles and their associated changes in unit co-firing within 25ms manifest within a small patch of cortex, and how spindles propagate in multiple patterns that are associated with distinct co-firing sequences that could support multiple patterns of plasticity within local networks. We moved several panels of former Fig.3 of the initial submission to the Supplementary Information since these results are not central to our thesis that conditions of plasticity are generated during spindles, but are still

important characterizations of the spindles we analyzed that may help with some of the interpretations.

In addition, several aspects appear inconsistent:

- The spatial dimension of spindles is not very well explored. Given the size of the Utah array and the different spatial scales in comparison to previous reports from ECoG electrodes, it seems likely that no independent spindles were quantified, but that the analysis mainly picks up one spindle source, where the spatial extent is predicted by the instantaneous amplitude of a single spindle source. How many 'independent' spindle events were captured at a given time or do most contacts simply measure 'one spindle source' - this question would have profound implications for the coherence results as reported here. Could the redundancy of spindles be quantified by e.g. ICA to test if the authors actually measure independent spindles? Bipolar referencing might also help in comparison to the utilized common average approach to isolate non-overlapping activity. Coherence values of close to 1 typically indicate volume-spread in the cortical tissue and not genuine connectivity.

Again, we thank the Reviewer for this important comment that we had failed to robustly discuss in our initial submission. Although spindles recorded by different contacts certainly could be due to volume conduction, and they would then have high coherence, our results as a whole are inconsistent with this interpretation. Conversely, our results support that at a given time, rather than there being multiple independent spindles, the detected spindles are coordinated and thus can indeed be considered part of the same phenomenon (i.e. the same spindle). Specifically, we found that for all distances there was a consistent spindle phase lag relationship as well as co-occurrence and coherence above chance. Spindles typically did not span the entire array (Supplementary Fig.2D) and there was a propensity for spindles to be detected on two channels simultaneously when those channels were close in proximity (Fig.4C), thus suggesting that spindles have some focality within the span of the Utah Array. We have updated the manuscript to explain that our results suggest unified spindling across contacts, but that this can be focal.

We detected spindles on individual channels because we wanted to be certain in our analysis that when analyzing spindling and/or co-firing during spindling at two sites there was indeed spindling detected at both sites. It is true that a spindle detected on a given contact may not necessarily be confined to that contact, and our results suggest that there is unified spindling across multiple contacts. Namely, we found that simultaneous spindling occurs preferentially at short distances, which is consistent with the conjecture that an individual spindle source was detected, and importantly phase lag and coherence between detected spindles was predicted by the distance between detected spindles (Fig.4B,D). We have updated the text so that it more clearly explains why spindles were detected on individual channels and that our results are consistent with unified spindling.

The critical observations arguing for a travelling wave of local generators are the strong linear relationship between spindle phase lag and distance, and the exactly corresponding relationship between the lag in co-firing and distance. While high coherence can be found between two contacts recording a single generator propagating to both by volume conduction, such conduction is effectively instantaneous and thus zero lag, in contrast to the spindle propagation velocity of $\sim 0.28\text{m/s}$ that we observed (Fig. 4B). Any remote possibility that this lag is artifactual is eliminated by the corresponding lag in coupled cell firing (Fig.3K-L).

Since several of the spatial findings reported in former Fig.3 of the initial version of the paper were not central to our thesis that conditions of plasticity are generated during spindles, we moved most of the panels of this figure to the Supplementary Information, since they still contain useful characterizations of the spindles that we analyzed.

- No metrics of directionality were used, so why do the authors report that spindle organize spiking activity? Wouldn't a more parsimonious explanation be that unit firing gives rise to the LFP phenomenon?

Thank you for pointing out this lapse of logic. LFPs represent synaptic and active transmembrane currents, which both cause, and are the result of firing; consequently, assigning directionality between them for a complex network phenomenon seems futile. We updated the text so that our wording is consistent with spindles being a unified LFP and spiking phenomenon.

- Why were spindle events that appeared in close proximity to delta events excluded? Delta-spindle coupling constitutes a hallmark of the systems memory consolidation theory. This seems relevant since spindles do not constitute a unitary phenomenon, but differ in their respective function depending on their coupling profile (see e.g. work by the Staresina lab).

We apologize for our lack of clarity; we did not exclude spindles that were detected in close proximity to delta events. Putative spindles were excluded if they coincided with large increases in 4-8 Hz or 18-25 Hz band power in order to exclude non-spindle events, such as theta bursts, that can extend into the spindle band. Furthermore, in this revision we detected downstates and upstates (please see updated Methods for details on detection, added Supplementary Fig.3 for example LFP and high gamma envelope means and associations with spindles, and Fig.2, Fig.3A-H, added Supplementary Fig.9 and 10 for results). As we reported earlier in this response, we found an increase in unit firing and co-firing during isolated spindles (those that did not occur in association with downstates or upstates) and an even greater increase during spindles that coincided with down-upstates. We updated the text to discuss the implications of this regarding systems memory consolidation theory, specifically in regard to how spindles that occur on the down-to-upstate transition may be particularly important for memory consolidation.

- The section 'Analysis of Spindle Propagation' lacks critical aspects of why certain steps were conducted and does not contain sufficient information to replicate any of present data.

We added details explaining why certain steps were done and provided the all necessary input parameters in order to replicate our work.

- Exact p-values should be reported, as well as the outcome of the primary test statistic and measures of effect size. In several instances the authors simply report $p < 0.05$, which is somewhat deprecated.

We added exact p-values and included t-statistics along with Cohen's d as a measure of effect size.

- Statistics: Did the authors account for the fact that different units came from different subjects or were all collapsed into one pseudo-population. This seems statistically invalid given that these are likely not independent observation.

We added results for individual patients for key tests in the paper (Supplementary Tables 1 and 2), which show that the results are reasonably consistent across patients.

- Estimating spindle coherence and connectivity in 25ms bins seems almost impossible given typical duration of a spindle of > 500 ms.

Coherence was always computed across the entire overlapping epoch of two spindles and not in 25ms bins. We updated the text so that this is clear.

- Recently, the Buzsaki lab reported (Gelinás et al 2016 Nature Med) that interictal discharges impact spindle generation, hence, it is necessary to provide some clinical information as well as a relationship of the observed effects to epileptic activity. This seems critical, since the authors report data that was obtained from tissue within the resection zone.

In order to address this, we performed an additional detection of inter-ictal spikes when the z-score of the analytic amplitude of the 20Hz highpass and the 200Hz highpass of the signal exceeded 7, and excluded a small percentage of spindles that occurred within ± 100 ms of these

events and also rejected baseline periods within ± 100 ms of these events. We visually examined events from each channel from each patient to validate the detections. We then re-ran all analyses on the clean spindles and baseline periods

Since Gelinas et al. (2016) found an average time offset from IED to spindle onset of 410ms, we generated a peri-IIS spindle onset histogram of all spindle events in our re-analyses for this revision, to confirm that there was no contamination with IIS (Supplementary Fig.2C). The distribution of spindle onsets was flat outside of the rejection zone indicating that the data used in our analyses did not contain IED-spindle coupling like that which Gelinas et al. (2016) found.

Fig 1: It should be made easier to see, which panels reflect single unit examples or which reflect group averages H) Why not show the mean firing of different neuron types as a function time relatively to the spindle peak?

We updated the figure legend to explain which panels reflect single unit examples versus averages. We added plots that show the mean firing of PY and IN as a function of time to spindle trough as Supplementary Fig.7.

Fig 3D/J: It is unclear how these Figures were created; both the text and legend lack the necessary details.

We updated both the Methods and the legend to explain how these figures were created.

Fig 4E/G/etc: Again, it is unclear what is on the y-axis and how these figures were created

We added details including what is on the y-axis and how the figures were created.

Taken together, the authors obtained a unique dataset and now have the opportunity to address some very timely questions in sleep physiology. However, it appears that this dataset was analyzed with a methods-centric perspective, but one misses a convincing demonstration that links observations and proposed mechanism. The main pitfall is that, in its present form, the title is misleading and that the data does not convincingly show that spindles indeed mediate plasticity.

REVIEWER COMMENTS

Reviewer #1 (Remarks to the Author):

The authors investigated the role of traveling sleep spindles for spike-timing-dependent plasticity (STDP) using electrocorticography and intracranial microelectrode recordings implanted into the superior or middle temporal gyrus of four patients with focal, pharmaco-resistant epilepsy. They found during sleep spindles an increased firing within and between cortical units (presumed pyramidal cells and/or interneurons, greatest within ~2 mm distance) during a time interval of 25 ms, necessary for STDP (as previously suggested in rodents). Spindles propagated in specific patterns at a velocity of ~0.23 m/s within a sub-centimeter scale, possibly facilitating plasticity across multiple networks during non-rapid eye movement (NREM) sleep.

The findings are highly interesting for the sleep and memory community, and the manuscript is well-written. However, data analyses appear to be not always described in sufficient detail. Please find my concerns listed below.

Major issues

- The criteria used for electrocorticography-based sleep staging are unclear. Please describe in more detail and/or cite respective literature. Does this electrocorticography-based sleep staging correspond to rules applied to EEG-based sleep staging (i.e., by Rechtschaffen & Kales or AASM)? How long were the "periods"/epochs used to classify sleep stages? How were other sleep stages such as stage 1 sleep or REM sleep classified? Please also provide a table indicating standard sleep architecture for each participant (i.e., total sleep time, time/percentage spent in stage 1, 2, 3 and REM sleep, wake after sleep onset, sleep efficiency, number of spindles, spindle density, number of epochs with potential epileptic seizures etc.).
- There is evidence that sleep spindles are different in patients with epilepsy, and this further depends on the type of epilepsy (e.g., Drake et al., 1991, Clin Electroencephalogr). Moreover, spindles may contribute to triggering nocturnal seizures (Montplaisir et al., 1981, Sleep). How can the authors rule out that the results are representing a general mechanism (that applies to healthy participants) that is not an atypical phenomenon due to epilepsy – especially, when the location of recording was strongly suspected and later resected? Did participants have epileptic seizures during recordings? How were interictal spikes identified and how was their occurrence related to the occurrence of spindles? etc.
- I am not convinced that the results on phase-locked unit spiking during spindles are robust, as they seem to be based on roughly 30 pyramidal cells and 20 interneurons only.
- Given the small number of patients, different ages, different probes, different location of probes, different hemispheres and gender differences, the authors should indicate in more detail if results were comparable between these parameters. For instance, in the results section ("Spindles are associated with an increase in unit spike rates"), the author indicated that there are similarities as well as differences for 1.0 mm and 1.5 mm probes, but they do not show any data or statistics (on the group level). It is further unclear why the authors used different silicon probes of different length and width for the four patients in the first place. For how long (hours/nights) were intracranial recordings performed in each patient? Were recording periods of similar duration for the four patients? How was the data averaged across the four patients in case of different recording lengths?
- How can the authors exclude that the signal of one presumed propagating spindle detected at different channels/locations does not actually correspond to two different spindles?
- Referencing needs to be updated. Mechanisms and functions of spindles have recently been comprehensively reviewed by Fernandez and Lüthi A (Physiol Rev 2020), and a review of spindles for memory consolidation has been provided by Klinzing J, et al. (Nat Neurosci 2019). I also miss ground laying early work by Timofeev/Steriade who were the first to propose a plasticity function ("augmenting response") of spindles as well as recent work (e.g. Niethard N et al, PNAS 2018) supporting this view. All this work should be integrated to provide a more balanced picture.

Minor issues

- p.15, last sentence: I do not follow the logic of this calculation. If 17% and 11% of neurons exhibiting phase preference were pyramidal cells and interneurons, respectively, what are the remaining 72%?
- Firing and co-firing during spindles might be generally increased during slow-oscillation up-states in slow-wave sleep. Did the authors also have the chance to look at spindle/slow oscillation coupling?
- In the Figure 1 caption for panel G, the authors state that they are showing raster plots of unit spiking of the "three" putative cell types, while in fact they are only showing raster plots for two cell types (PY, IN). Please adapt.
- The definition of "shuff-spindles" (sleep spindles with shuffled spikes) should be introduced at first mention in the main text.
- Typo in Discussion, third paragraph: should read "STDP".

Reviewer #2 (Remarks to the Author):

Dickey et al. analyzed the spatiotemporal patterns of putative spindle activity during sleep (NREM2-3) and associated single unit firing recorded from the human brain using Utah arrays. They explore a valuable dataset with a large number of simultaneously recorded single units. The main claim of the paper is that spindles synchronize single neurons at short cortical distances, enabling the plasticity changes. This finding could contribute to our understanding of brain plasticity during sleep, possibly related to memory consolidation. However, the proportion of pyramidal cell pairs with ordered co-firing during spindles seems to be <2% (Table 2), which raises the possibility that it might happen by chance. In addition, several methodological issues require careful clarification before the results could be properly interpreted and conclusions evaluated.

1) Data inclusion: The recordings were performed continuously for 4-21 days/subject, and one dataset/subject was selected for the analysis. Was the selection based on pre-established objective criteria, or was it random? Were any datasets excluded at this stage? It is unclear if the data was collected during overnight sleep, sleep periods during the day, or both. If the data was collected during overnight sleep, did the authors include in the analysis all the NREM2-3 periods recorded during the same night? If not, what was the basis for selection?

2) Single unit isolation quality and stability: It is unclear what isolation quality criteria were used to select single units. The only hint is given on page 7, noting that single units were required to have refractory periods of ' ≥ 1 ms'. The refractory period is 3 ms and the percentage of spikes during this period is an indication of single unit contamination by spikes from other units, which is a criterion for unit selection. The authors should show detailed unit quality metrics and use the inclusion criteria widely accepted in human single unit literature (e.g. Kaminski et al., 2020; Supplementary Figure 2). Finally, for recordings lasting several hours or overnight, neuronal drift might affect the results, so the authors should use temporal stability as the criteria for unit inclusion.

3) Removal of epileptic activity: The authors stated that all the detected spindles were visually inspected for the absence of epileptic activity (Page 8). However, the number of detected spindles is ~500000 (Page 12), which doesn't seem feasible for human visual inspection. Even if the spindles were visually inspected, what about the non-spindle periods used for baseline?

4) Spindle detection: Spindles are defined across the rodent and primate literature as lasting 0.5-2/3 sec, and the authors note this in the introduction (Page 2). However, criteria for spindle detection used in this paper include the events as short as 200 ms, with the mean duration ~400 ms. This suggests that the large proportion of analyzed events are as short as 2-3 oscillatory cycles, which raises the possibility of false positives. For these results to be comparable with other

literature (e.g. Nir et al., 2011; Andrillon et al., 2011), the authors should use the widely accepted spindle detection criteria.

5) Rigor and reproducibility: The authors reported a large increase in single unit firing during spindles, contrary to Andrillon et al. (2011). The baseline for this comparison is defined as periods outside of the spindle epochs, likely include cortical down-states, a period during which most of the neurons are silent by definition (see the raster plots in Fig.1G). To demonstrate the firing rate increases during spindles, down-state periods should be detected based on accepted criteria and excluded from the baseline. This detail is also critical for other comparisons that use non-spindle baselines.

6) Spindle coherence: Coherence is a connectivity measure well known to be inflated by the oscillation amplitude. To account for this, the authors should either use the power-balanced design or a connectivity measure less influenced by amplitude (e.g. phase-locking value).

7) When describing the correction for multiple comparisons, the authors should define the basis for corrections in each case and provide the exact n.

8) Spike time tiling procedure is not described in detail.

9) When shuffling procedures are used to generate null distributions, the process should be precisely described (what is being shuffled? what shuffling method is used? within what time range?).

Reviewer #3 (Remarks to the Author):

Review of Dickey et al.

In the present manuscript, the authors report data from 4 epilepsy patients who were undergoing presurgical evaluation with implanted electrodes. Here additional Utah arrays were implanted during invasive Phase-II monitoring. The authors focused on NREM sleep and report both field potentials as well as spiking data from putative single neurons. The work builds on a series of recent papers by the same group and is well-grounded in the literature.

Overall, the authors report a series of analyses that focus either on (a) LFP-spike locking, (b) spike correlations, (c) spatial correlations and trajectories of traveling waves. There are no pre-post sleep comparisons, no behavioral task and no quantification if any plasticity processes occurred. While the paper reflects an impressive tour-de-force using a variety of methods, the main conclusion that traveling spindle waves create the necessary conditions for long-term plasticity is not supported by the present data.

The key issue of the present manuscript is that multiple analyses are being introduced taking advantage of a unique dataset, which is very difficult to obtain, but it lacks theory and a clear motivation. Some of the analyses appear rather arbitrary and only loosely connected. In the end, the reader is left with an observation of how firing changes during spindles, without further quantification. Then the spatial dimension is introduced, several correlations are reported, then the traveling waves idea is introduced and in the end the results are interpreted as reflecting a neurophysiological scenario that mediates STDP. The authors might want to highlight the observation, which supports the plasticity aspect.

In addition, several aspects appear inconsistent:

- The spatial dimension of spindles is not very well explored. Given the size of the Utah array and

the different spatial scales in comparison to previous reports from ECoG electrodes, it seems likely that no independent spindles were quantified, but that the analysis mainly picks up one spindle source, where the spatial extent is predicted by the instantaneous amplitude of a single spindle source. How many 'independent' spindle events were captured at a given time or do most contacts simply measure 'one spindle source' - this question would have profound implications for the coherence results as reported here. Could the redundancy of spindles be quantified by e.g. ICA to test if the authors actually measure independent spindles? Bipolar referencing might also help in comparison to the utilized common average approach to isolate non-overlapping activity. Coherence values of close to 1 typically indicate volume-spread in the cortical tissue and not genuine connectivity.

- No metrics of directionality were used, so why do the authors report that spindle organize spiking activity? Wouldn't a more parsimonious explanation be that unit firing gives rise to the LFP phenomenon?

- Why were spindle events that appeared in close proximity to delta events excluded? Delta-spindle coupling constitutes a hallmark of the systems memory consolidation theory. This seems relevant since spindles do not constitute a unitary phenomenon, but differ in their respective function depending on their coupling profile (see e.g. work by the Staresina lab).

- The section 'Analysis of Spindle Propagation' lacks critical aspects of why certain steps were conducted and does not contain sufficient information to replicate any of present data.

- Exact p-values should be reported, as well as the outcome of the primary test statistic and measures of effect size. In several instances the authors simply report $p < 0.05$, which is somewhat deprecated.

- Statistics: Did the authors account for the fact that different units came from different subjects or were all collapsed into one pseudo-population. This seems statistically invalid given that these are likely not independent observation.

- Estimating spindle coherence and connectivity in 25ms bins seems almost impossible given typical duration of a spindle of > 500 ms.

- Recently, the Buzsaki lab reported (Gelinas et al 2016 Nature Med) that interictal discharges impact spindle generation, hence, it is necessary to provide some clinical information as well as a relationship of the observed effects to epileptic activity. This seems critical, since the authors report data that was obtained from tissue within the resection zone.

Fig 1: It should be made easier to see, which panels reflect single unit examples or which reflect group averages H) Why not show the mean firing of different neuron types as a function time relatively to the spindle peak?

Fig 3D/J: It is unclear how these Figures were created; both the text and legend lack the necessary details.

Fig 4E/G/etc: Again, it is unclear what is on the y-axis and how these figures were created

Taken together, the authors obtained a unique dataset and now have the opportunity to address some very timely questions in sleep physiology. However, it appears that this dataset was analyzed with a methods-centric perspective, but one misses a convincing demonstration that links observations and proposed mechanism. The main pitfall is that, in its present form, the title is misleading and that the data does not convincingly show that spindles indeed mediate plasticity.

We thank the reviewers for their additional valuable feedback on our manuscript. Below we address each comment and are also submitting an updated version of the manuscript and supplementary information. We look forward to further consideration of our work.

Reviewer #1 (Remarks to the Author):

Overall, the authors have very carefully and satisfactorily addressed all of my comments. Two minor points remain:

We thank the reviewer for their feedback and have addressed these points below.

1. The authors determined sleep stages based on standard criteria by Rechtschaffen and Kales (1968). Based on these criteria, NREM sleep is divided into S1 to S4. Spindles are typical for S2-S4. However, it appears that the authors have now based their analyses only on S2 and S3. It is unclear if and why S4 has not been included. Please, clarify and make sure that S4 is included.

We apologize for not adequately referencing our method. Rechtschaffen and Kales has been updated by the AASM as described in Silber et al. In the new standard clinical criteria, the sleep stage previously labeled NREM period S2 is now labelled N2, and periods S3 and S4 are now combined into N3. This is the standard we followed and we have updated the manuscript accordingly.

2. In the rebuttal letter, the authors indicate that “all patients who had a Utah Array implanted into tissue that was determined to be epileptogenic were excluded from the study”. However, it is unclear if and how many patients had to be excluded based on this criterion. This information needs to be added to the ms.

Four patients out of a total of 10 were selected for this study based on the presence of high quality units, typical sleep periods, and the absence of significant epileptiform activity.

Reviewer #2 (Remarks to the Author):

The authors have performed extensive additional analysis, but they have not addressed several issues adequately. I am elaborating on each point below, with the authors' rebuttal text. Overall, it is not clear from the presented results if the co-firing during spindles is inducing neural plasticity or that it has relevance for memory consolidation. First, plasticity is by definition accompanied by the connectivity weight changes. The authors did not attempt to demonstrate the change in co-firing statistics as a function of co-firing history, which would resemble the plasticity-induced weight changes. Second, there is no evidence that the described neuronal dynamics during spindles are selective for neuronal pairs involved in memory consolidation, as no memory task was performed. With these deficiencies and the methodological problems listed below, the manuscript, in my opinion, is not suitable for publication in the current state.

We thank the reviewer for their detailed explanations of their remaining concerns, and we hope to have adequately addressed these concerns below.

Our results show that spindles create conditions necessary for plasticity, but it is true that we do not show definitively that there are synaptic weight changes, or that the units that are modulated during spindles are necessarily involved in memory consolidation since we do not have memory

task data from these patients with these rarely implanted microelectrodes. We updated the Discussion of our paper to explain these limitations.

We would also like to note that our results show that a substantial population of neurons, even within the small region we sampled, have increased, organized co-firing during spindles. Based on the relationship between co-firing and distance, as well as the previously reported neuron densities in human anterior temporal lobe (DeFelipe et al., 2002), we estimated that the number of layer III PY that co-fire more during spindles with a given PY within a radius of 4mm, beyond what would be expected from a simple increase in firing rate, was 37,532 (see Methods).

1) Determination of chance level.

(6% of all pairs significant with 0.1% expected by chance for $\alpha=0.001$)

(with 5% expected by chance for $\alpha=0.05$)

What is the basis for calculating chance levels and why is it different in these two cases?

Chance levels simply correspond to the definition of alpha. These were different statistical tests and both used commonly accepted alpha values. For the former test, we could have used $\alpha=0.05$ instead of $\alpha=0.001$ leading to an even higher proportion of significant results, but we chose a lower value since there were many comparisons and multiple different types of tests used in conjunction where significance at a lower α would be more convincing.

2) Unit stability criteria.

We also would like to note that none of our results would be expected to be caused by unit temporal instability.

For the recordings lasting for several hours, neuronal drift is quite common, and it could affect the results presented in the manuscript. For example, suppose the two neurons decrease firing in parallel due to neuronal drift. In that case, the co-firing statistics could be artificially inflated due to the fact that BOTH neurons don't fire during the later spindle episodes. Therefore, the authors need to use the stability criteria to either exclude the units with unstable firing from the analysis or limit the analysis of a given pair to the period when both units were showing stable firing.

The unit pair co-firing statistics in our paper evaluate co-firing during 1) spindles vs. non-spindles and 2) spindles with actual spike times vs. spindles with shuffled spike times. For a given unit pair, if one or both units decreased spiking over time due to drift then it would not result in increased amounts of co-firing during spindles compared to non-spindles (there would be decreased co-firing for both) or during spindles with actual spike times vs. shuffled spike times (organized co-firing would not increase). In other words, the shuffling occurs within spindles and so any possible drift over time will apply equally to both organized and control firing rates. This is now clarified in the text.

3) Spindle detection criteria. The authors justified using a very short lower spindle duration threshold by claiming that intracranially recorded spindles are very short duration events and offer the previous paper from their lab as the evidence for acceptance of these criteria (Hagler et al., 2018). However, they are probably well aware that the most cited literature on intracranially recorded spindles in humans was based on 500 msec lower threshold (Nir et al.,

2011; Andrillon et al., 2011), as well as the vast literature from rodents. The histogram of spindle durations indicates that most detected events were shorter than the standard in the field. Since the authors claim that there is no difference between the spindles shorter and longer than 500 msec, they should demonstrate their results using the broadly accepted spindle duration definitions (>500 msec.)?)

We continue to offer two responses to this concern. First, there is no firm lower bound for intracortical spindle duration. In addition to Hagler et al, we would note that Sritharan et al., *J Neurophysiology*, 2020, also found, using cortical microelectrode recordings, a very similar spindle duration distribution (Sritharan Fig.1G, copied below) as ours (Dickey, Supplementary Fig.4A, copied below). Their distribution extended below 300 ms and up to ~1200 ms, with the peak of the distribution close to 300 ms, and the majority of spindles <500 ms, quite similar to ours, as is shown in these plots from the two papers:

Sritharan et al., *J Neurophysiology*, 2020

Dickey et al., Supplementary Fig.4A

Second, using a different spindle duration threshold does not change the basic findings of the paper. For example, we found that there was a significant increase in the percent of baseline spike rate of units in both the >500 and <500ms duration groups, and furthermore the size of this increase did not change significantly between the two groups (Supplementary Fig.4, copied below).

Supplementary Figure 4: Unit spiking during shorter compared to longer spindles. A-B, spike rates during concatenated spindle epochs as log-percent of baseline spike rate during shorter (<500ms) spindles (A) and longer (>500ms) spindles (B). Baseline spike rate was computed as the spike rate during concatenated NREM epochs between spindles. Color circles shows the mean firing rate of one unit. Black horizontal mark shows mean and vertical shows SEM. Dashed horizontal line shows baseline spike rate (100%). The mean percent of baseline spike rate for PY during shorter spindles was $219.16 \pm 14.22\%$ and longer spindles was $234.55 \pm 20.68\%$. These were both significant increases from baseline ($p_{PY,short} = 7e-22$, $p_{PY,long} = 3e-21$, Bonferroni-corrected $\alpha = 0.025$ for 2 spindle types, one-sample two-sided Wilcoxon signed-rank test, $z_{PY,short} = 9.62$, $z_{PY,long} = 9.45$). The mean percent of baseline spike rate for IN during shorter spindles was $160.16 \pm 10.46\%$ and longer spindles was 164.36 ± 11.41 . These were also both significant increases from baseline ($p_{IN,short} = 4e-6$, $p_{IN,long} = 2e-6$, Bonferroni-corrected $\alpha = 0.025$ for 2 spindle types, one sample two-sided Wilcoxon signed-rank test, $z_{IN,short} = 4.61$, $z_{IN,long} = 4.71$). There was no significant difference between PY for short vs. long spindles or IN for short vs. long spindles ($p_{PY} = 0.40$, $p_{IN} = 0.18$, Bonferroni-corrected $\alpha = 0.025$ for 2 unit types, paired two-sided Wilcoxon signed-rank test, $z_{PY} = 0.84$, $z_{IN} = -1.34$).

Therefore, the data support that for cortical spindles recorded intracranially with microelectrode probes, those that are 300-499 ms come from the same general population as those that are >500 ms.

3) Connectivity determination.

"Coherence is sensitive to the covariance, or joint variance, or how much the amplitude of two signals changes together, although it is not sensitive to the amplitudes of the individual signals since in computing the coherence the variance of both signals is divided out. We propose that a purely phase-based measure, such as phase-locking value, has less biological relevance since co-amplitude is relevant to the underlying neuronal activity. Two small signals can have significant phase-locking, but this is not necessarily biologically meaningful.

It has been demonstrated that the signal amplitude can affect the coherence estimate (Srinath and Ray, J of Neurophys, 2014):

“Importantly, even when phases were completely randomized between two electrodes, amplitude correlations introduced significant coherence. To quantify the contributions of phase consistency and amplitude correlations to coherence, we simulated pairs of sinusoids with varying phase consistency and amplitude correlations. These simulations confirmed that amplitude correlations can significantly bias coherence measurements, resulting in either over- or underestimation of true phase coherence.”

In addition, if the authors are making a case that the directional co-firing of a neuron pair is essential for plasticity induction and neurons are phase-locked to local spindles, it would be expected that the spindles on the electrodes where those neurons are recorded show consistent phase relation; hence, the most relevant connectivity measure would be phase-based. If the authors are concerned that low-amplitude events would contaminate phase-based connectivity estimates, they could simply use the appropriate amplitude threshold. Using the standard duration threshold for detecting spindle events (500 msec, as mentioned above), would also help avoid low amplitude events.

Indeed, in our paper, we have used both phase-based and coherence-based analyses to evaluate the relationships between spindles and co-firing. With regard to phase-based measures, our results show that co-firing is locked to spindle phase (Fig.3I,J) and that there is a strong linear relationship between the phase lag between co-occurring spindles detected at two sites and co-firing between co-located units (Fig.3K,L).

We acknowledge in the text (page 31, line 947-949) that coherence is affected both by phase and by signal co-amplitudes. Nonetheless, it is a useful and very commonly used metric for quantifying the relationship between two time-series. We share the concern of the reviewer that coherence between two very small signals may be of questionable biological significance but that concern would not apply in our study because we required spindles in both locations. Specifically, in the detection of spindles, we require that each spindle has a large increase in spindle band amplitude (but not adjacent frequency bands). In all of our coherence analyses we require that spindles are co-occurring at both sites. Therefore, our coherence analyses are restricted to locations and times for which there are co-occurring large amplitude spindles.

To confirm that varying the amplitude of only one of two signals does not affect coherence between these signals, we ran a test on simulated data. We generated two 0.5 s duration 12 Hz sine waves with sampling rate of 1000 Hz and applied a random phase offset ranging from 0 to 0.5π at each sample to one of the signals. The coherence values were then computed and were equal when the signal amplitudes were the same or one when was amplified (e.g. by a factor of 1.2, 3, 10, 100, etc.).

Since coherence is a widely used measure to assess the correlation of two biological signals, and since we have also shown a link between spindle phase and co-firing, we believe that we have sufficiently addressed this concern.

Reviewer #3 (Remarks to the Author):

The authors address all comments in great detail. I also found their responses to the other reviewers comments valuable. I have no additional comments. The manuscript likely constitutes a valuable resource for future studies. Therefore, it'd be desirable if at least some of the data

could be made available to replicate how firing patterns change relative to field potentials. In terms of interpretation, it should be made sufficiently clear that all data stems of anterior lateral temporal cortex, which is not part of the network that is typically associated with sleep-dependent memory consolidation (since the authors rest the hypothesis on theoretical work as reviewed in e.g. Klinzing 2019, which very much focuses on the mPFC-MTL pathway).

We thank the reviewer for their feedback.

We will make the figure source data publicly available. We also would like to see the raw data as available as possible to investigators outside our group, and indeed have provided it in the past upon request, resulting in the following publications:

1. Dehghani N, Peyrache A, Telenczuk B, Le Van Quyen M, Halgren E, Cash SS, Hatsopoulos NG, Destexhe A. Dynamic balance of excitation and inhibition in human and monkey neocortex. *Scientific reports*. 2016;6.
2. Teleńczuk B, Dehghani N, Le Van Quyen M, Cash SS, Halgren E, Hatsopoulos NG, Destexhe A. Local field potentials primarily reflect inhibitory neuron activity in human and monkey cortex. *Scientific reports*. 2017;7:40211.
3. Le Van Quyen M, Muller LE, Telenczuk B, Halgren E, Cash S, Hatsopoulos NG, Dehghani N, Destexhe A. High-frequency oscillations in human and monkey neocortex during the wake–sleep cycle. *Proceedings of the National Academy of Sciences*. 2016;113(33):9363-8.
4. Peyrache A, Dehghani N, Eskandar EN, Madsen JR, Anderson WS, Donoghue JA, Hochberg LR, Halgren E, Cash SS, Destexhe A. Spatiotemporal dynamics of neocortical excitation and inhibition during human sleep. *Proc Natl Acad Sci U S A*. 2012;109(5):1731-6. Epub 2012/02/07. doi: 10.1073/pnas.1109895109.
5. Dehghani N, Bedard C, Cash SS, Halgren E, Destexhe A. Comparative power spectral analysis of simultaneous electroencephalographic and magnetoencephalographic recordings in humans suggests non-resistive extracellular media. *J Comput Neurosci*. 2010;29(3):405-21. Epub 2010/08/11. doi: 10.1007/s10827-010-0263-2.

However, thus far we have been unable to place the raw data in a public repository. These data were collected over a long period of time, and involved different IRBs, scientists, and clinicians. The data sharing agreement currently does not permit making the data publicly available but we are working on obtaining the necessary permissions and agreements.

We have updated the Discussion to make it clear that all data come from anterolateral temporal cortex and that this region is not typically associated with sleep-dependent memory consolidation in rodents. However, rodents have no clear homologue to anterolateral temporal cortex, which in humans is considered to be a key part of the memory system (e.g., Vaz et al. 2020; Vaz et al., 2019).

REVIEWERS' COMMENTS

Reviewer #1 (Remarks to the Author):

The authors have carefully and satisfactorily addressed all of my comments. This is an interesting paper ready to be published.

One minor point: The term "down-upstate" in the abstract ("... especially those co-occurring with down-upstates") is probably confusing to the broader readership and should be replaced by a more meaningful phrasing (e.g., down-to-upstate transition).

Reviewer #2 (Remarks to the Author):

With clear caveats that the authors acknowledge (the lack of memory task and no attempt to measure the plasticity effects), the paper is ready for publication. I appreciate the authors' diligence in addressing my concerns.

Reviewer #3 (Remarks to the Author):

The authors addressed all queries in detail, I do not have any additional comments or concerns.

Response to Reviewers

Reviewer #1 (Remarks to the Author):

The authors have carefully and satisfactorily addressed all of my comments. This is an interesting paper ready to be published.

One minor point: The term "down-upstate" in the abstract ("... especially those co-occurring with down-upstates") is probably confusing to the broader readership and should be replaced by a more meaningful phrasing (e.g., down-to-upstate transition).

We thank the reviewer for raising this point of possible confusion and have updated this term to "down-to-upstate transitions" in the abstract as well as made our down-to-upstate terminology more clear throughout the rest of the manuscript.

Reviewer #2 (Remarks to the Author):

With clear caveats that the authors acknowledge (the lack of memory task and no attempt to measure the plasticity effects), the paper is ready for publication. I appreciate the authors' diligence in addressing my concerns.

We thank the reviewer for their help to clarify the limitations of this study. We appreciate their feedback.

Reviewer #3 (Remarks to the Author):

The authors addressed all queries in detail, I do not have any additional comments or concerns.

We thank the reviewer for their valuable input.